# A flanking-nicks prime editor (FLICK-PE) system to boost prime editing in dicots

Mengyan Bai [1,4], Jieping Zhang [1,4], Wenxin Lin [2,4], Yufan Zhou [1], Mengmeng Jiang [1], Haijie Wu [1], Chunyan Peng [3], Jieni Lin[1], Fanghui He [1], Huaqin Kuang[3] & Yuefeng Guan [1] ✉

Prime editing (PE) enables precise genome modifications to mammalian cells and monocot staple crops, but remains relatively challenging in dicot plants. Here, we develop a Flanking-Nicks Prime Editor (FLICK-PE) system that boosts editing efficiency in soybean and tobacco. We show that optimization for PE by adding a nicking sgRNA could dramatically enhance intended-editing efficiency in soybean. Inspired by this observation, we design a FLICK-PE strategy to confer a pair of nicks flanking the target site. In soybean, FLICK-PE achieves on average a 15.7-fold increase in intended-editing efficiency compared to PE2, and a 2.2-fold increase compared to PE3. Using FLICK-PE, we efficiently engineer glyphosate resistance in soybean by introducing TAP–IVS mutations in *EPSPS1a*, achieving three amino-acid substitutions and an intended editing efficiency of 21.1%. This approach yields stable edited soybean varieties with vigorous glyphosate tolerance and minimal growth penalties in a field trial. FLICK-PE also demonstrates efficacy in tobacco, underscoring its broad applicability and versatility for rapid, precision breeding in agriculturally vital crops.

Precision gene editing is desirable for addressing modern agricultural challenges through targeted crop-improvement strategies[1]. Derived from the CRISPR–Cas system, prime editing (PE) emerges as a transformative genome-engineering technology enabling precise base substitutions, insertions, and deletions[2–4]. This system combines a Cas9 nickase with a reverse transcriptase (RT) to directly integrate edits guided by the prime-editing guide RNA (pegRNA), containing a primer binding site (PBS) and an RT template (RTT), into target genomic loci[2].

Since its inception, substantial efforts focus on optimizing PE efficiency in plants[4]. The optimizations include, but are not limited to, enhancements in RT thermostability and activity[5–9], pegRNA secondary-structure engineering[10–15], integration of conditional excision systems[16], introducing surrogate system[17–19], and chromatin opening through heterologously expressing the human RNA m[6]A demethylase hFTO[20]. In monocot species such as rice[10,12,14,21–24] and maize[11,15,21], PE efficiencies (up to 90% for intended edits) are achieved for specific editing scenarios, enabling applications like multiplex PE[8,19], large fragment manipulation[25], and kilobase-scale DNA integration[26]. However, unlike mammalian systems wherein optimized PE tools enable intended insertions or deletions of larger gene fragments, scarless insertion of larger fragments remains a key challenge in plants, reflecting the current limitations alongside progress in the plant PE field.

PE implementation in dicot plants has been more challenging than in monocots, with editing efficiencies low and persistent difficulties in generating heritable edits[27–29]. Breakthroughs in tomato and Arabidopsis demonstrate success through combinatorial optimizations, including strong, constitutive-promoter-driven expression of PE components, engineered epegRNA architectures, geminiviral replicon-mediated delivery, and post-transfection heat treatments[30]. Through

[1]Guangdong Provincial Key Laboratory of Plant Adaptation and Molecular Design, Innovative Center of Molecular Genetics and Evolution, School of Life Sciences, Guangzhou University, Guangzhou, China. [2]Sanya Institute of China Agricultural University, Sanya, China. [3]College of Life Sciences, Fujian Agriculture and Forestry University, Fuzhou, China. [4]These authors contributed equally: Mengyan Bai, Jieping Zhang, Wenxin Lin. ✉e-mail: guan@gzhu.edu.cn

advanced optimization strategies, precise insertion of a 10-bp heat-responsive element in the promoter of an invertase gene has been achieved in tomato and rice[25,31]. Nevertheless, these advances remain insufficient for routine application in many major dicot crops, such as the legume soybean (*Glycine max*). Besides leveraging canonical strategies from mammalian systems and monocotyledons, additional optimization approaches may be required to enhance PE efficiency in dicots.

Emerging evidence suggests that manipulating the DNA mismatch-repair (MMR) pathway could increase PE efficiency[32]. In rice, over-expression of the dominant-negative mutant *MLH1dn* or conditional suppression of *MLH1*, encoding an MMR component, enhances PE outcomes[11,15,16]. The PE3 strategy introduces additional nicks on the non-editing strand, which may guide MMR to use the editing strand as a template for repair, and PE3 shows a 2–4-fold improvement over PE2 in human cells[2,32]. In rice, PE3 shows a 1–3-fold increase over PE2, which is considered associated with the targeting activity and position of the nicking sgRNA[10]. In dicots, PE3 efficiency varies from uninheritable to over 10% intended-editing rate in different studies, yet a systemic comparison with PE2 is lacking[18,31,33]. Thus, the effectiveness of DNA nicking in plant genomes varies, and is potentially related to positional effects of different nicking sgRNAs, expression efficiency of the nicking sgRNA, or MMR preferences may differ at different editing sites across different taxa[10]. So far, it remains inconclusive whether nicking is effective for PE in plants, particularly in dicot species.

Soybean is a globally vital staple crop for plant protein and oil, with its demand expected to continue to rise. However, the efficiency of gene-editing technology in soybean lags behind staple monocot crops. Heritable PE has not been realized in soybean, despite its importance and numerous efforts on PE optimization. In this study, we perform extensive canonical optimization on PE-component expression, yet achieve only a low PE editing efficiency that is barely inherited in stably transformed lines. We find that nicking sgRNAs enhance soybean PE efficiency by 4.7–11.8-fold over PE2, and we develop the Flanking-Nicks Prime Editor (FLICK-PE) system, introducing two nicking sgRNAs flanking the target site on the non-editing strand. This strategy further improves PE efficiency in soybean and tobacco, with an average 15.7-fold and 2.2-fold higher intended-editing rate than PE2 and PE3, respectively. Using FLICK-PE tools, we generate inheritable, edited soybean lines bearing a tripartite amino-acid change with glyphosate resistance, and further demonstrate FLICK-PE utility in tobacco.

## Results

### Canonical optimization of PE2 tools achieves poor editing efficiency in soybean

We first tested combinations of known optimization strategies for feasibility as soybean PE tools (namely SoyPE-V1 to SoyPE-V5), including: (1) driving *nCas9* (H840A) variants from the soybean *M4* promoter (proM4) for efficient editor expression[34]; (2) using nCas9−MMLV PEmax with stronger single-strand cleavage capability and nuclear localization[8,10,12]; (3) testing soybean strong promoters (proU6 composite[11], proM8L and proUBQ3) for enhancing pegRNA transcription; (4) using a single transcript-unit configuration driven by proM4 to allow synergistic expression of *nCas9−RT* and pegRNAs[20] (Fig. 1a and Supplementary Fig. 1a). We then designed PE configurations and pegRNAs targeting seven target loci in the soybean c.v. William82 genome and tested their functionality in the first instance using transgenic hairy roots (Fig. 1b and Supplementary Fig. 2). By Hi-TOM deep sequencing[35], efficiency of intended prime edits was rather low for all tested tools, with up to 19.52% intended-edit reads of total sequencing reads in each transgenic event, and none of these PE tools produced detectable editing at target 7 (Fig. 1c). Among these tools, SoyPE-V5 (proM4 driving *PEmax* and proUBQ3 driving the pegRNA) showed relatively high efficiency, the highest percentages of intended edits at six loci were 6.23–19.52% (Fig. 1c). SoyPE-V3 with proU6 composite driving the

pegRNA showed similar yet slightly lower intended PE efficiency with SoyPE-V5 (Fig. 1c).

We previously showed that altering chromatin by co-expressing a human hFTO as an assister could increase PE efficiency in rice and knock-out editing efficiency in soybean[20]. Here, we tested the effect of hFTO in SoyPE-V5. With its co-expression (SoyPE-V6), intended PE efficiency was increased at four of seven target sites, in comparison with SoyPE-V5, yet is still at a rather low level (<19.78% intended-editing reads) (Fig. 1d and Supplementary Fig. 1a).

We next performed stable transformation of soybean c.v. HC-6 with SoyPE-V6 targeting the *EPSPS1a* locus (encoding 5-enolpyruvylshikimate-3-phosphate (EPSPS) synthase in the shikimate pathway that biosynthesizes aromatic amino acids and is targeted by glyphosate) to introduce triple amino-acid substitution mutations T183I/A184V/P187S (TAP–IVS) that confer glyphosate resistance in *Amaranthus hybridus*[36] (designated target-site 1 in hairy-root PE testing). Among 41 stable T₀ transgenic plants, we only found 15 lines with no more than 5% of reads with intended PE, which are not heritable (Fig. 1e).

In conclusion, integrating canonical strategies—including optimized PE protein configuration, enhanced expression of PE transcripts, and chromatin modulation—led to detectable PE events in transgenic soybean hairy roots. However, these approaches remain insufficient to achieve heritable PE edits in stable-transgenic soybean.

### Nicking sgRNAs dramatically improves prime-editing efficiency in soybean

We postulated two potential reasons for inefficient PE in soybean. First, the expression of PE elements may not be as sufficient as the regular CRISPR−Cas9 system. Second, DNA-repair mechanisms may prefer the non-edited sequences to the RT-editing strands, making PE editing difficult. We attempted several strategies to test these hypotheses.

We first compared the expression level of PE elements with regular CRISPR−Cas9 elements in transgenic events. The CRISPR−Cas9 vector pGES401 could achieve an average of 50–80% indel editing ratios in transgenic hairy roots[20]. qRT-PCR indicated that *nCas9* and pegRNA expression from SoyPE-V5 in transgenic roots was at a similar level to *Cas9* and the sgRNA in hairy roots edited by pGES401 (Supplementary Fig. 3). Thus, the expression level of PE components does not appear to bottleneck PE efficiency in soybean.

Since PE components can be highly expressed in soybean, but PE efficiency is very low, we hypothesized that the editing strand cannot be effectively integrated in the process of PE editing, thereby limiting efficiency (Fig. 2a). We tested whether additional nicking can lift efficiency. The effectiveness of the nicking sgRNA at the non-editing strand in PE3 (with an additional nicking sgRNA at the non-editing strand) was demonstrated in mammalian cells[2], yet remains variable across different reports in plants[6,10,37–39], which might be related to positional effects of nicking. We thus compared the effectiveness of PE3 with PE2 in soybean (Fig. 2b). Next, we tested whether competition from the original sequence at the editing strand (known as a 5′ flap) might limit the integration of the newly synthesized RT strand. We therefore designed a strategy in which an additional nick downstream of the targeted site at the editing strand was introduced, possibly enhancing the cleavage of the 5′ flap (designated PE−Editing-Strand nick, or PE−ES^nick) (Fig. 2c).

We designed multiple nicking sgRNAs for seven targets using the SoyPE-V5 vector. For each target, we designed 1–3 nicking sgRNAs for both PE−ES^nick and PE3. The positions of nicking sgRNAs were located from −60 to 85-bp downstream of the target sites (Supplementary Fig. 2). Overall, PE−ES^nick showed minor effects on PE efficiency compared with PE2, except at target 7 (Fig. 2d–j and Supplementary Fig. 4). On the contrary, we were surprised to see that certain PE3 strategies could enhance efficiency. The average intended-editing percentage of total reads per transgenic event

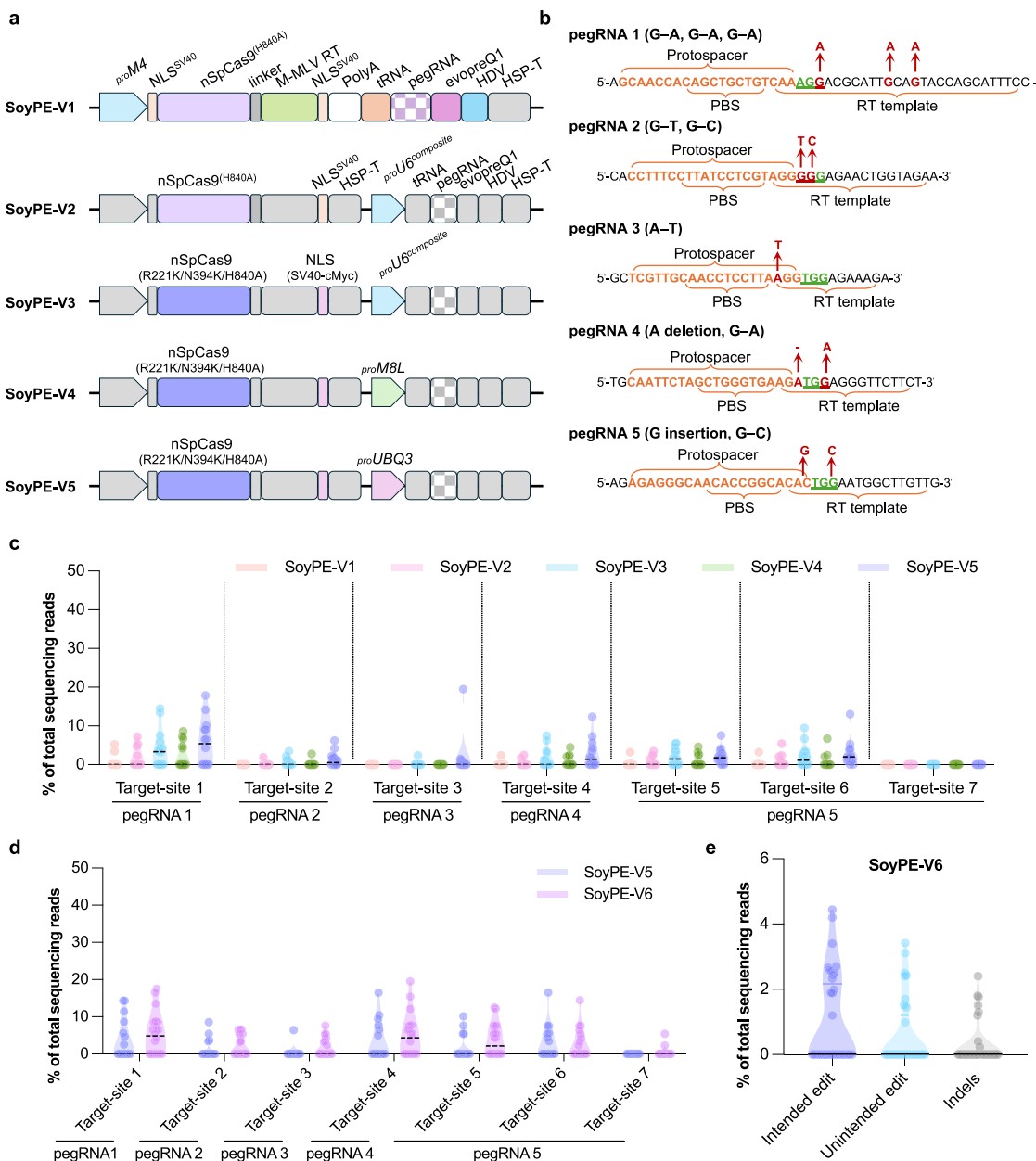

**Fig. 1 | Canonical optimization of PE2 tools in soybean. a** Vector structure of SoyPE-V1 to SoyPE-V5. **b** Design of pegRNA-1 to pegRNA-5 for targeting seven soybean genes. Red arrows indicate the intended base change. **c** Percentage of intended editing efficiency at seven loci induced by SoyPE-V1 to SoyPE-V5 in transgenic hairy roots and assayed by Hi-TOM ($n = 15$). **d** Percentage of intended editing efficiency at seven loci induced by SoyPE-V5 and SoyPE-V6 in transgenic hairy roots and assayed by Hi-TOM ($n = 20$). **e** Percentage of intended editing, unintended editing, and indel efficiency at *EPSPS1a* (target 1) induced by SoyPE-V6 in independent stable transgenic lines assayed by Hi-TOM ($n = 42$). All data points are shown on the plots. The central black lines in the violin plots represent the median in (**c**–**e**). The horizontal gray lines represent the lower and upper quartiles, and the shaded areas represent data-distribution density (**c**–**e**). Source data are provided as a Source data file.

gave 4.7–11.8-fold increases over non-nick PE2 systems (Fig. 2k). For example, at target-site 1, the PE3 vector with a nick (+65 bp) gave a statistically significant 11.8-fold increase in average intended-editing efficiency compared to the PE2 vector, achieving the highest intended-editing rate of 54.9% (Fig. 2d). At target-site 2, the PE3 vector with a nick (+27 bp) achieved intended editing with the highest rate reaching 76.4%, whereas the highest intended-editing rate in PE2 was only 10.7% (Fig. 2e). At target-site 3, PE3 vectors with nicks at +47 bp exhibited a 8.2-fold increase in average intended-editing rate compared to the PE2 vector, achieving an intended-editing rate of 79% (Fig. 2f). At target-site 4, PE3 vectors with a nick (+57 bp) achieved an intended-editing rate of 65.2% and a 6-fold

increase compared to than PE2 (Fig. 2g). For pegRNA-5, which targets three homologous genes, PE3 vectors carrying different nicking sgRNAs achieved effective editing in all three genes. In contrast, the PE2 vector failed to produce effective editing—with PE3 showing a 5.1–5.8-fold average increase in intended-editing rate at target sites 5–7. Meanwhile, a control vector with a nontarget nick (wherein the nicking sgRNA sequence contains mutations and fails to target the genes) also did not induce effective editing (Fig. 2h–j). Effective-editing efficiency (calculated as the proportion of hairy roots with intended-editing ratio > 20% by Hi-TOM deep sequencing among all tested roots) of the PE3 strategy showed that the effective intended-editing efficiency ranged from 8.33–34.9% across the seven target

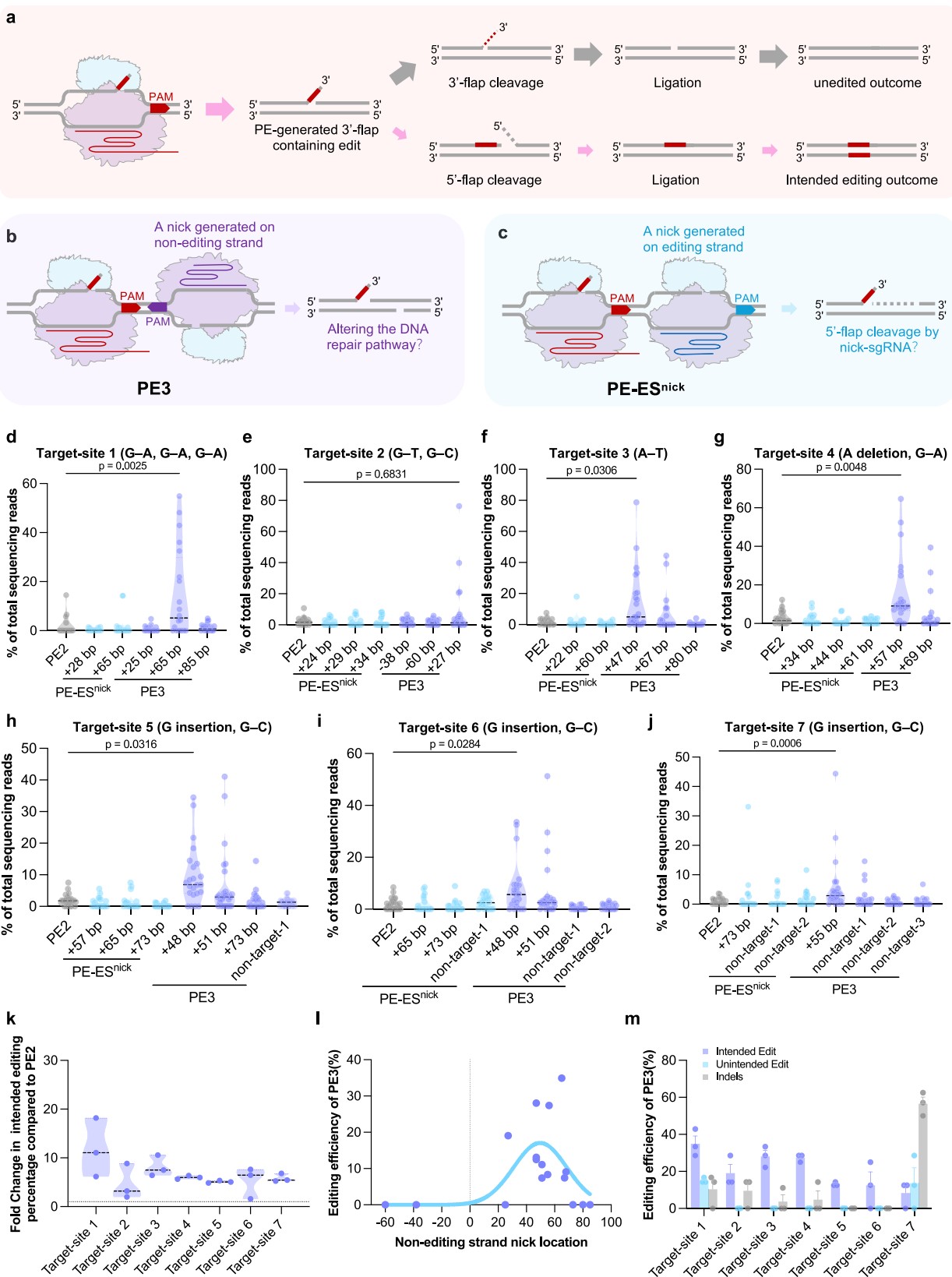

genes, with varying levels of unintended editing and indels (Fig. 2m). In contrast, the PE2 vectors failed to induce effective editing at all target genes (Fig. 2d–j).

In all efficient editing events mediated by PE3, unintended edits and indels were moderately increased, except for target-site 7 (Supplementary Fig. 4). The most-effective nicking sgRNAs were designed at positions 40–70-bp downstream of targets, consistent with the principle proposed in mammalian cells[2] (Fig. 2l). To validate further the high-frequency intended-editing events identified by Hi-TOM sequencing, Sanger sequencing found peaks in samples with Hi-TOM-identified high intended-editing rates exceeding 20% (Supplementary Fig. 5). In conclusion, nicking sgRNAs as part of a PE3 strategy

**Fig. 2 | Evaluation of prime-editing efficiency for PE3 and PE–ES^nick strategies.** **a**–**c** Overview of the mechanisms by which PE3 and PE–ES^nick strategies. **a** If the 3′ flap of the editing strand fails to be efficiently integrated, the genome reverts to wild type (upper panel). Conversely, successful integration achieves intended prime editing (lower panel). **b** PE3 strategy adds a nick downstream of the target site, targeting the non-editing strand. **c** PE–ES^nick strategy adds a nick downstream of the target site that targets the editing strand. **d**–**j** Percentage of intended editing efficiency at seven target loci induced by PE3 and PE–ES^nick in hairy roots determined by Hi-TOM. The "nontarget" vector is defined as follows: it uses a nicking sgRNA designed to introduce multiple SNPs at the target site, preventing it from

generating effective nicks; "nontarget-1", "nontarget-2" and "nontarget-3" refer to variants whereby the nicking sgRNAs differ in the positions of the introduced SNPs. All three serve as control vectors for the PE3 vector. **k** Fold change in intended editing efficiency of the most-efficient vector in PE3 compared to PE2 across seven target sites. **l** Correlation between the position of nick sgRNA and intended editing efficiency in PE3. **m** Editing efficiency of the PE3 strategy, with error bars representing the standard deviation (SD) ($n = 20$ in (**d**–**g**), $n = 27$ in (**h**–**j**)). Data in (**d**–**j**) were analyzed by two-tailed Mann–Whitney testing ($P < 0.05$). Source data are provided as a Source data file.

achieved efficient editing in soybean hairy roots, whereas the PE2 strategy failed to produce effective editing.

## FLICK-PE achieves efficient prime editing in soybean via dual nicking

Given the dramatic improvement in PE3 efficiency achieved through nicking of the non-editing strand, we hypothesized that the limited PE efficacy in soybean or other dicots might be related to a preference for MMR to use the non-editing strand over the RT-editing strand as a repair template. Consequently, manipulating the DNA repair pathway on the non-editing strand is key. We wondered if further destabilization of the non-editing strand by dual nicking might further improve PE efficiency in soybean. To test this, we designed three strategies: type I was designed as flanking nicks, so that the two nicking sgRNAs were located upstream and downstream of the target site at the non-editing strand, respectively, designated Flanking-Nick Prime Editor (FLICK-PE); Type II (asymmetric nicking) was designed to produce asymmetric nicks with both nicking sgRNAs located downstream of the targets on the non-editing strand. Type III (cross-strand nicking) double-strand nicks incur one nick at the editing strand and another at the non-editing strand (Fig. 3a). All nicks were designed within 20–85 bp surrounding the targets (Supplementary Fig. 2).

We first selected target sites 3 and 5 to compare these three strategies (Supplementary Fig. 2c, e). At both sites, FLICK-PE achieved more effective intended editing compared with PE3, although it is accompanied by a comparable rate of unintended editing and indels (Fig. 3b–d and Supplementary Fig. 6). In contrast, Type-II and -III strategies only increased unintended editing and indels, with unchanged or reduced intended-editing efficiency (Fig. 3b–d).

Considering FLICK-PE showed robust editing and maintained specificity, we chose this strategy for further evaluation using five target sites, including target-site 1 (previously tested with PE3) and four new targets (target sites 8–11) (Supplementary Fig. 7). Compared to PE2, FLICK-PE demonstrated higher intended-editing efficiency at all five targets (Fig. 3e–i). Specifically, the average intended-editing percentage of total reads per transgenic event showed 15.7-fold (maximum 35.8-fold) increases over PE2 (Fig. 3j). Compared to PE3, FLICK-PE gave an average 2.2-fold increase in intended-editing ratios across the five tested target sites (Fig. 3j). At target-site 11, PE2 and PE3 failed to induce any effective editing, whereas FLICK-PE achieved successful intended editing at 33.64% (Fig. 3i). Effective intended-editing efficiency, defined as the proportion of transgenic roots with an intended-editing ratio >20% (as determined by Hi-TOM deep sequencing) relative to all tested roots, exhibited a 1.7-fold improvement across all seven tested targets compared to PE3, with an overall average effective intended-editing efficiency of 21% (vs. 12.7% for PE3). Specifically, the average effective intended-editing efficiency at target-site 1 increased from 27.3% in PE3 to 40.9% in FLICK-PE (Fig. 3k). PE2 still showed no effective editing across all new targets. (Fig. 3k). Further, we examined 105 predicted off-target sites for all pegRNAs and nicking sgRNAs tested by FLICK-PE in the hairy root system using Hi-TOM deep sequencing. The read ratio of Indels/SNPs detected at these predicted off-target sites was <20% (consistent with the threshold for effective

intended edits, with 20% serving as the threshold for defining effective editing), and at most sites, the detected read ratio of Indels/SNPs was around 5% or lower. However, at the predicted off-target sites for pegRNAs 1 and 8, the highest read ratios of Indels/SNPs detected were 8.52% and 8.98%, respectively. Therefore, there remains a certain degree of off-target risk at some predicted off-target sites. (Supplementary Figs. 8–14).

We also tested the effectiveness of manipulating MMR by co-expressing an *GmMLH*1 RNAi construct with PE2/3 and FLICK-PE (Supplementary Fig. 15a) in transgenic hairy-root composite plants. The average percentage of intended editing relative to total reads per hairy root at two target sites for PE2-*MLH1i* and PE3-*MLH1i* was 2.3/2.4-fold and 1.6/1.3-fold that of PE2/PE3 alone, respectively. This result indicated that, like in rice, repressing *MLH* expression also improves PE efficiency in PE2 and to a lesser extend in PE3[16]. However, FLICK-PE-*MLH1i* failed to improve the efficiency in compared with FLICK-PE (Supplementary Fig. 15b–g). Meanwhile, the effective intended-editing efficiency of FLICK-PE at the two target sites (40.3 and 26.5%) remains higher than that of both PE3 (23.1 and 18.3%) and PE3-*MLH1i* (30.2 and 23.4%) at the same sites.

In summary, FLICK-PE boosts the efficiency of precise PE in soybean while preserving specificity, apparently through relieving the inhibitory effect of the MMR pathway.

## FLICK-PE generates heritable edits that confer glyphosate resistance in the field for an elite soybean cultivar

To validate the efficiency of FLICK-PE in soybean stable transformation, we tested three target sites (target-sites 1, 3, and 5) in the elite cultivar HC-6, using PE2 and PE3 as controls. In $T_0$ transformants, all positive transformants were analyzed via Hi-TOM deep sequencing (intended-editing reads accounting for >20% were defined as "effective intended editing", which were mostly heritable). PE2 failed to generate effective editing at any of the three tested target sites following stable transformation. In contrast, FLICK-PE generated the highest effective intended editing at all three tested target sites. At target-site 1, Hi-TOM identified four lines carrying intended edits from FLICK-PE, with an intended-editing efficiency of 21.1% (4/19 lines), whereas PE3 achieved an effective intended-editing efficiency of 8.7% (2/23 lines). Meanwhile, a certain proportion of indels (2/19 lines) were observed using FLICK-PE (Fig. 4a). At target-site 3, FLICK-PE gave an effective intended-editing efficiency of 11.1% (3/27), accompanied by 7.4% (2/27) unintended edits and 3.7% (1/27) indels (Fig. 4b). PE3 reached 6.7% (2/30) effective intended editing. Consistent results were observed at Target-site 5, with FLICK-PE achieving an effective intended-editing efficiency of 16% (4/25), compared to 3.1% (1/32) for PE3 (Fig. 4c). Collectively, FLICK-PE enables effective intended editing using soybean stable transformation, with an efficiency range of 11.1–21.1% (mean 16.1%), higher than that of PE3 (average efficiency 6.2%, range 3.1–8.7%) (Fig. 4d). Consistent with Hi-TOM results, all the above effective intended edits were detectable by Sanger sequencing (Fig. 4e–g). We also evaluated potential off-target effects for all pegRNAs and nicking sgRNAs via Hi-TOM deep sequencing. and found that the read ratio of Indels/SNPs at most off-target sites ranged from 0 to 5%. However, in individual samples, the read ratio at certain off-target sites reached

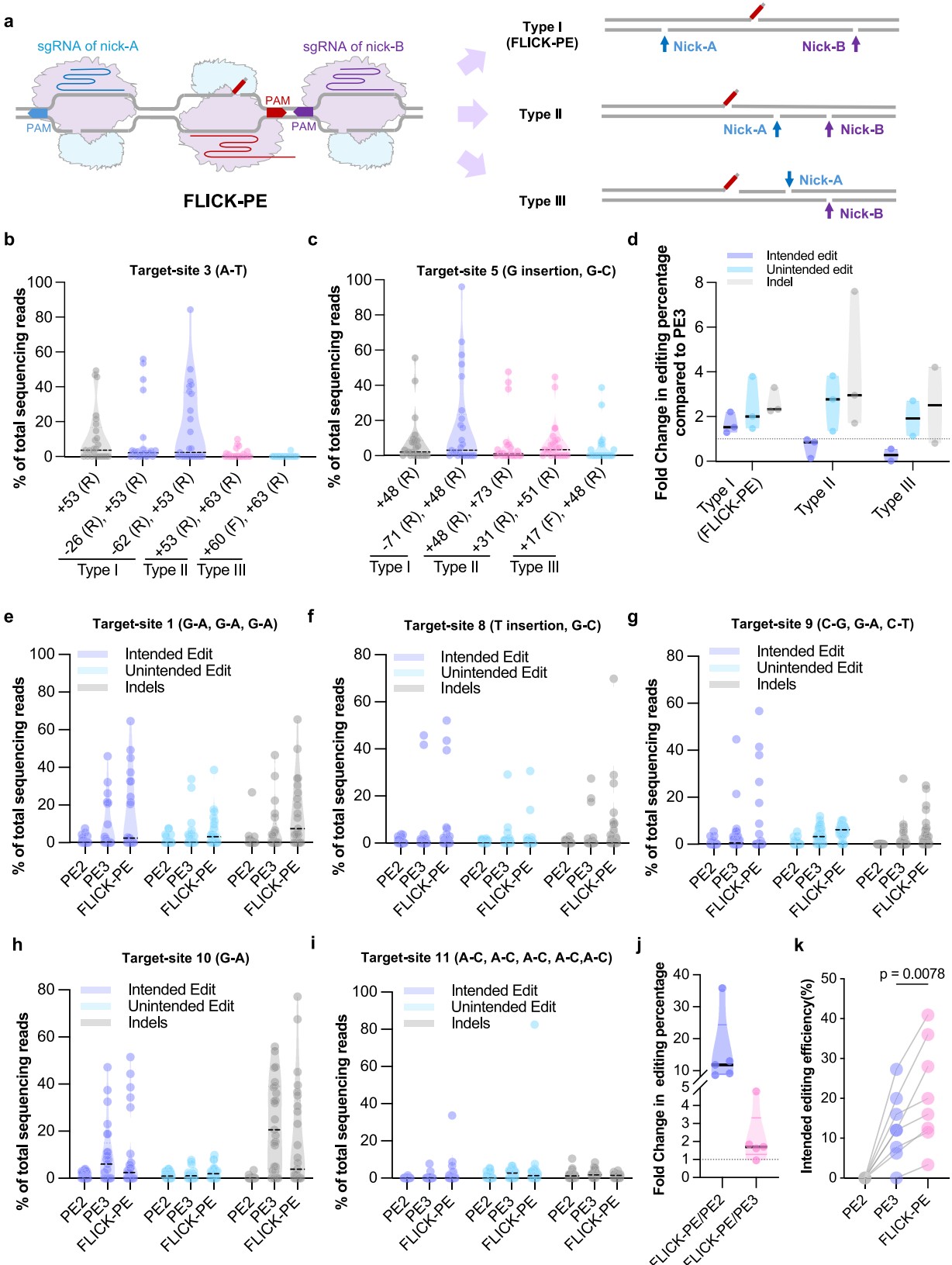

7.53%, higher than that in the control group. Nevertheless, the Indels/SNPs at all sites were below the 20% threshold for effective editing, consistent with the results in hairy roots. While no obvious off-target effects were observed (the read ratio of Indels/SNPs å 20%), there remains a certain risk of off-target events that could generate additional Indels/SNPs (Supplementary Fig. 16).

We evaluated hereditability of FLICK-PE-mediated editing in T$_1$ progenies by screening edited plants targeting target-site 1, which introduced the TAP–IVS polymorphisms at *EPSPS1a*[36](Fig. 5a). Indeed, we obtained heritable and homozygous edited plants (genotype: *epsps*1$a^{TAP–IVS}$) in the T$_1$ progenies, confirming that FLICK-PE enables heritable gene editing via stable soybean transformation (Fig. 5b, c).

**Fig. 3 | Developing FLICK-PE strategy in soybean. a** Overview of dual-nicking PE strategy. Percentage of intended-editing efficiency at target 3 (**b**) and target 5 (**c**). Numbers on x-axis labels indicate the distance (in base pairs) between nick sgRNA cleavage sites and the pegRNA-induced nick. Plus and minus symbols designate downstream or upstream positioning relative to the pegRNA cleavage sites, respectively. The letters F and R represent the sgRNA targeted editing-strand or non-editing strand, respectively. **d** Fold change in intended editing, unintended editing and indel percentages for all three dual-nicking strategies compared to PE3 across targets 3 and 5 ($n = 25$ in **b**–**d**). Percentage of intended editing, unintended

editing and indels at target 1 ($n = 22$) (**e**), target 8 ($n = 26$) (**f**), target 9 ($n = 32$) (**g**), target 10 ($n = 25$) (**h**) and target 11 ($n = 30$) (**i**) induced by FLICK-PE, PE3 and PE2. All the data points are shown on the plots. **j** Fold change in intended-editing percentage for FLICK-PE versus PE2 and FLICK-PE versus PE3 (targets 1, 8–11). **k** Effective intended-editing efficiency of PE2, PE3 and FLICK-PE. The two points linked by a line denote the consistent design of the pegRNA and the first nick sgRNA positioned downstream of the target site. Data in (**j**, **k**) is from same sample size as in (**b**–**i**). Data in (**k**) were analyzed by the two-tailed Wilcoxon matched-pairs signed rank test ($P < 0.05$). Source data are provided as a Source data file.

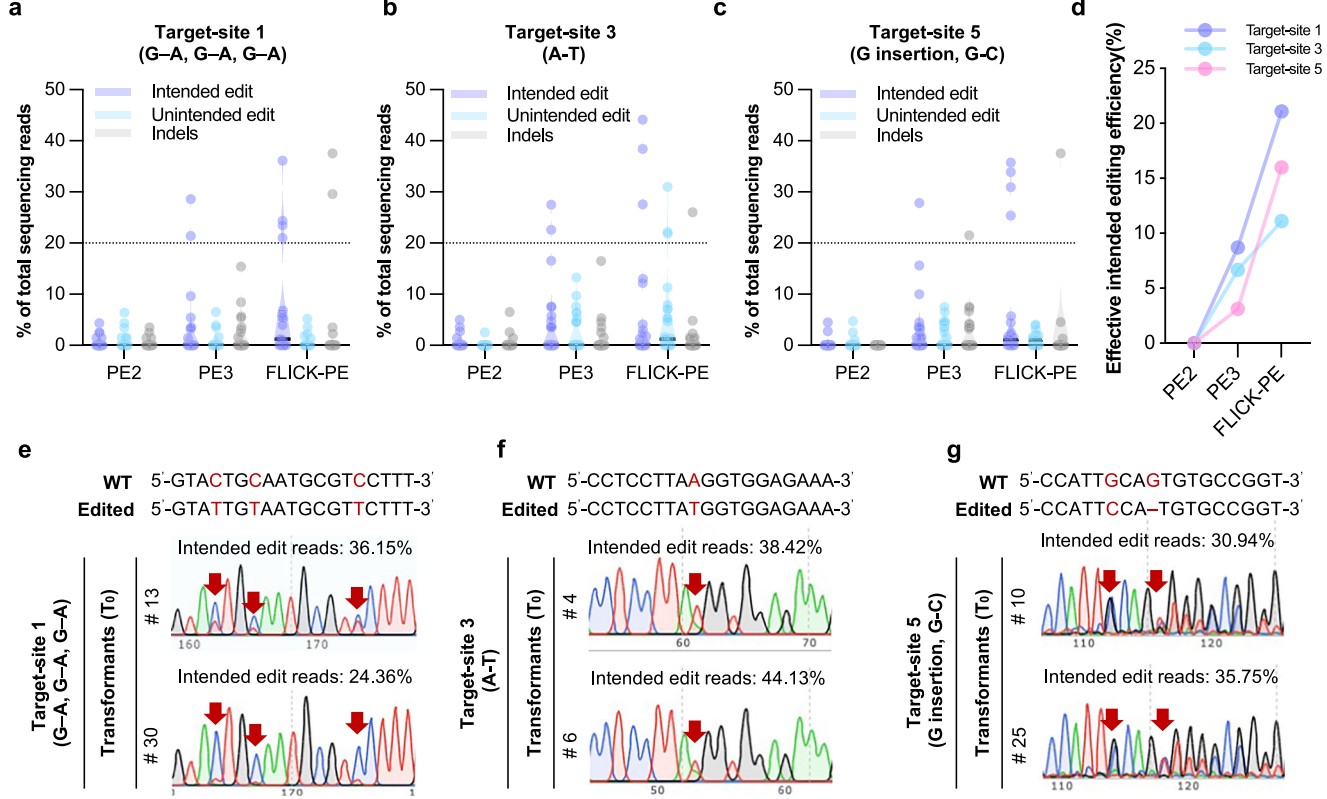

**Fig. 4 | FLICK-PE mediates efficient intended editing in soybean stable transformation.** Percentages of intended editing, unintended editing and indels at target-site 1 (PE2 $n = 25$, PE3 $n = 23$, FLICK-PE $n = 19$) (**a**), target-site 3 (PE2 $n = 21$, PE3 $n = 30$, FLICK-PE $n = 27$) (**b**) and target-site 5 (PE2 $n = 27$, PE3 $n = 32$, FLICK-PE $n = 25$) (**c**) induced in $T_0$ soybean stable transformants by PE2, PE3 and FLICK-PE, as determined via Hi-TOM deep sequencing. **d** The effective intended-editing efficiency of PE2, PE3 and FLICK-PE in $T_0$ soybean stable transformants. The effective intended-editing events were defined as Hi-TOM deep sequencing-validated

intended editing rates $\geq 20\%$. Sanger sequencing chromatograms of two $T_0$ lines carrying effective intended edits (indicated by red arrowheads) at target-site 1 (**e**), target-site 3 (**f**) and target-site 5 (**g**) induced by FLICK-PE. The central black lines in the violin plots represent the median in (**a**–**c**). The horizontal gray lines represent the lower and upper quartiles, and the shaded areas represent data-distribution density (**a**–**c**). The "intended edit reads" above the chromatograms indicate the proportion of intended edits in this line detected via Hi-TOM sequencing (**e**–**g**). Source data are provided as a Source data file.

Homozygous $epsps1a^{TAP–IVS}$ soybean plants were fertile, and development was only slightly affected (but without statistical significance) for plant height, pod number, seed-number per plant, yield per plant, or 100-seed weight (Fig. 5d–i). *EPSPS1b* was not targeted (the most-likely potential off-target site for pegRNA-1) and showed no off-target editing at all 15 potential off-target sites (Supplementary Figs. 16a and 17).

We tested glyphosate resistance for homozygous $epsps1a^{TAP–IVS}$ plants without the editing transgene at four herbicide concentrations: 1× (1.23 kg a.i. ha⁻¹), 2× (2.46 kg a.i. ha⁻¹), 4× (4.92 kg a.i. ha⁻¹), and 8× (9.84 kg a.i. ha⁻¹) recommended working concentrations of glyphosate. Wild-type HC-6 control plants died at all concentrations 10 day post-spraying (applied 14 days after germination). In contrast, $epsps1a^{TAP–IVS}$ plants showed no obvious adverse effects at 2× concentration. At higher doses, leaf curling and desiccation symptoms were observed in edited plants, but these remained alive and fertile

(Fig. 5j). In conclusion, we successfully used FLICK-PE to generate new soybean germplasm that confers resistance to high doses of glyphosate, without compromising plant fertility.

## FLICK-PE confers efficient intended editing in tobacco

To confirm whether FLICK-PE has broad utility for other dicotyledonous plants, we conducted testing in tobacco *Nicotiana benthamiana* (*N. benthamiana*) Solanaceae. We first engineered a tobacco-optimized PE vector system, whereby the 35S-enhanced promoter drives the expression of an nCas9–MMLV PEmax fusion protein. The *AtU*6 composite promoter[11] was employed to drive pegRNA and nicking sgRNA expression, and an integrated eGFP marker was used for efficient transgenic screening by fluorescence (Fig. 6a and Supplementary Fig. 1b). Six combinations of pegRNAs and different nicking sgRNAs targeting four loci were designed, covering small-fragment insertions, deletions, substitutions, and combinations thereof (Fig. 6b

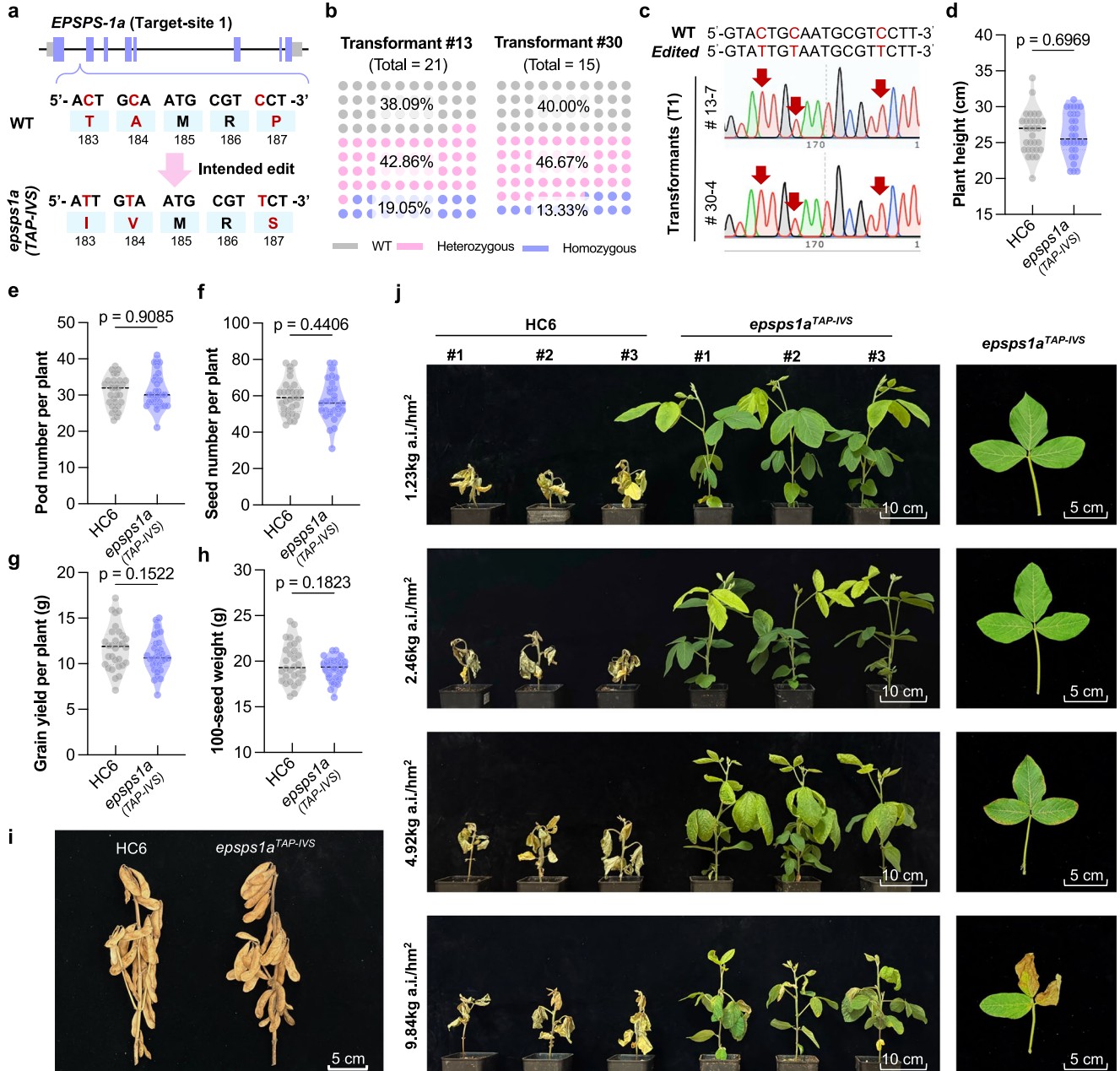

**Fig. 5 | FLICK-PE generates glyphosate-resistant soybean. a** *EPSPS1a* genomic structure and mutation sites that result in the *epsps1a*^TAP-IVS genotype. **b** Summary of heritable mutations after precision editing in two T₁ lines. **c** Sanger sequencing chromatograms of T₁ lines carrying homozygous *TAP–IVS* edits to *EPSPS1a*. **d–i** Plant height (**d**), pod number per plant (**e**), seed number per plant (**f**), grain yield per plant (**g**) and 100-seed weight (**h**) of wild-type c.v. HC-6 and an *epsps1a*^TAP-IVS edited line after harvest (*n* = 30). Two-tailed Student's *t* tests (*P* < 0.05) were performed in (**d–h**). **i** Phenotypes of mature HC-6 and *epsps1a*^TAP-IVS plants grown in the field (Guangzhou, 2024). **j** Responses of HC-6 and *epsps1a*^TAP-IVS plants to glyphosate spraying at four concentrations (1.23 kg a.i. ha⁻¹, 2.46 kg a.i. ha⁻¹, 4.92 kg a.i. ha⁻¹, and 9.84 kg a.i. ha⁻¹) administered 14 days after germination (under controlled conditions: 400 µmol m⁻² s⁻¹ photosynthetically active radiation, 13 h light (26 °C)/11 h dark (24 °C) photoperiod, and 65% relative humidity.) Left: shoot phenotypes. Right: representative *epsps1a*^TAP-IVS leaves. Plants were photographed 10 days after treatment. The central black lines in the violin plots represent the median in (**d–h**). The horizontal gray lines represent the lower and upper quartiles, and the shaded areas represent data-distribution density (**d–h**). Source data are provided as a Source data file.

and Supplementary Fig. 18), and functionality was tested by transient transformation of leaf epidermal cells.

Overall, FLICK-PE consistently improved intended-editing efficiency across all tested targets. At target-site 1 with two vectors designed (FLICK-PE-1 and FLICK-PE-2), their intended-editing efficiency was higher than that of their corresponding PE2 and PE3 vectors, reaching 16.51% and 11.97%, respectively, representing a 4.71–6.49-fold increase over PE2 and 1.21–1.96-fold increase over PE3, as confirmed by Hi-TOM analysis of DNA samples extracted from GFP-positive leaf regions (Fig. 6c). At target-site 2, the intended-editing efficiency of FLICK-PE in both sets of vectors was also higher than that of PE2 and PE3, at 17.38% and 17.77%, with improvements of 9.21–9.43-fold (compared to PE2) and 1.13–1.77 fold (compared to PE3), respectively (Fig. 6d). At target-site 3, FLICK-PE achieved an intended-editing efficiency of 37.24%, a 5.67-fold increase compared to PE2. (Fig. 6e). At target-site 4, FLICK-PE intended-editing efficiency (8.02%) was higher than that of PE3 (4.54%), and meanwhile, it showed a 15.27-fold improvement compared to PE2. (Fig. 6f). In all six tested strategies,

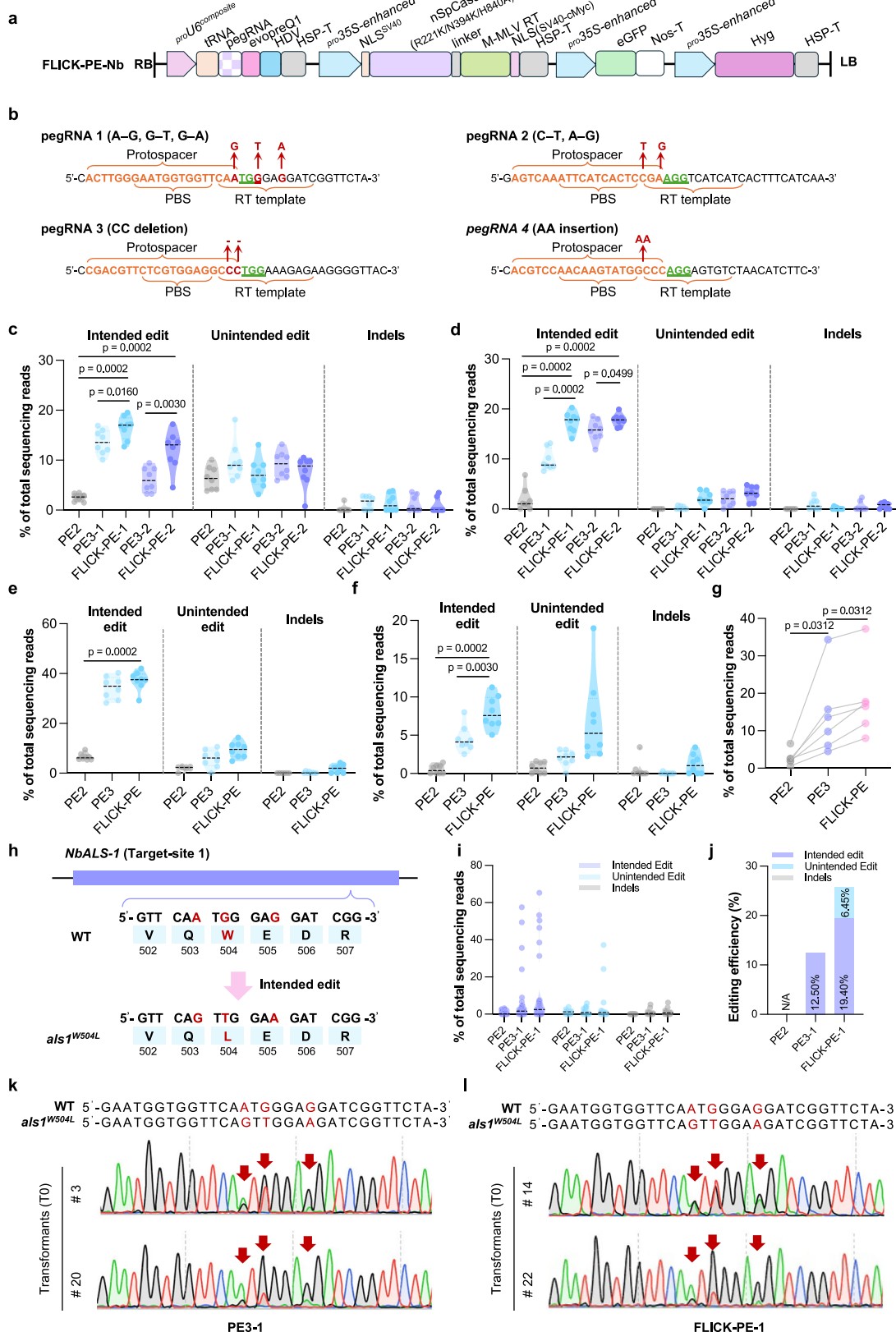

FLICK-PE outperformed both PE2 and PE3. Compared to PE2, FLICK-PE demonstrated an average 8.15-fold improvement, with the maximum improvement reaching 15.27-fold. Meanwhile, compared to PE3, FLICK-PE achieved an average 1.49-fold improvement (Fig. 6g).

In subsequent tests using stable transformation, tobacco *ALS1* was selected for precise editing (target-site 1 in the transient transfection system) (Fig. 6h). Through this approach, we successfully endowed the W504L amino-acid substitution on ALS1 (Fig. 6i). Hi-TOM sequencing showed that PE2 failed to achieve precise editing, whereas FLICK-PE achieved an editing efficiency of 19.4%, surpassing that of PE3 (12.5%) (Fig. 6j). Sanger sequencing further confirmed that samples with effective editing had double peaks in the chromatograms (Fig. 6k, l).

**Fig. 6 | FLICK-PE enables precision editing in *N. benthamiana*. a** FLICK-PE vector structure designed for *N. benthamiana*. **b** Design of pegRNA-1 to pegRNA-4 targeting four genomic loci. Red arrows indicate the intended base change. Percentage of intended editing, unintended editing, and indels at target 1 (**c**), target 2 (**d**), target 3 (**e**) and target 4 (**f**) induced by PE2, PE3 and FLICK-PE in tobacco cells. PE3-1 and PE3-2 differ in the nick sgRNAs used; FLICK-PE-1 and FLICK-PE-2 differ in their two nick sgRNAs, with PE3-1 and FLICK-PE-1 forming one control vector pair, and PE3-2 and FLICK-PE-2 forming another (**c**, **d**). All the data points are shown on the plots. *n* = 8. **g** Comparison of average precision-editing percentages of PE2, PE3 and FLICK-PE across all tested target sites. **h** *NbALS*-1 genomic structure and mutation sites that confer $als1^{W504L}$. **i** Percentage of intended editing, unintended editing and indels at *NbALS*1 (target-site 1) induced by PE2, PE3 and FLICK-PE in tobacco $T_0$ stable transformants (PE2 *n* = 32, PE3 *n* = 32, FLICK-PE *n* = 31). **j** Effective efficiency of PE2, PE3 and FLICK-PE in tobacco $T_0$ stable transformants. Sanger sequencing chromatograms of two $T_0$ lines carrying intended editing (indicated by red arrowheads) in *ALS*1 induced by PE3 (**k**) and FLICK-PE (**l**). Data in (**c**–**f**) were analyzed by two-tailed Mann–Whitney testing ($P < 0.05$). Data in (**g**) were analyzed by two-tailed Wilcoxon matched-pairs signed rank testing ($P < 0.05$). Source data are provided as a Source data file.

In conclusion, FLICK-PE achieves high-efficiency, precise editing in both transient transformation systems and stable transformation systems in tobacco. Verified by Hi-TOM deep sequencing, its editing efficiency outperforms PE2 and PE3 strategies. Following transient transformation of tobacco leaves with all six FLICK-PE vectors and stable transformation with the FLICK-PE vector targeting *ALS*1, the average intended-editing efficiency reached 18.3% and 19.4%, respectively.

## Discussion

Here, we report FLICK-PE for efficient PE in dicot plants. FLICK-PE, compared to canonical PE strategies[4], is unique in that it uses two nicking sgRNAs flanking the targets in the non-editing strand. In soybean hairy roots, intended editing increased by an average of 15.7-fold (up to 35.8-fold) compared to PE2 and 2.2-fold (up to 4.8-fold) compared to PE3 (Fig. 3j). Subsequent validation in stable-transformation systems confirmed its efficiency, with precise editing efficiencies of 11.1–21.1% in soybean $T_0$ lines and 19.4% for tobacco lines (Figs. 4 and 5). Using FLICK-PE, we obtained gene-edited glyphosate-resistant soybean lines in an elite cultivar background with normal growth (Fig. 5). FLICK-PE broke efficiency barriers in dicot plants and achieves high editing efficiencies for heritable, intended prime edits.

In dicots, canonical optimizations (e.g., engineered epegRNAs, improved PE-component expression, viral delivery, heat treatments) enabled intended editing in tomato and *Arabidopsis*[30,31], yet these strategies remain inadequate for routine use in dicots. This suggests that endogenous factors beyond PE-component optimizations may bottleneck efficiency in dicots. Here, robust improvement imparted by FLICK-PE indicates that the preference of edited and non-editing strands in MMR may be crucial for PE in dicot plants. Nicks on the non-editing strand theoretically could steer MMR toward using the editing strand as a template for repair, leading to intended PE[32]. We found that the flanking-nicking sgRNAs were efficient and maintained specificity, while cross-strand nicking sgRNAs, both designed to bind downstream edited and non-editing strands, increased the indel rate. An explanation is that cross-strand nicking sgRNAs may increase the risk of double-stranded breaks, yet a mechanistic understanding remains elusive. RNAi interference against *MLH*1 could improve PE2/PE3 efficiency in soybean, but showed little effects on FLICK-PE activity. Both nicking sgRNAs and *MLH* perturbation may work through inhibiting the possibility of repair of precise-editing products[32,40]. Thus, FLICK-PE with two nicking sgRNAs might override the effect of *MLH*1 RNAi. Considering that *MLH*1 perturbation causes sterility in stable-transgenic rice[16], FLICK-PE represents a more feasible and effective approach in soybean than *MLH*1 RNAi.

Contemporary strategies used in mammalian cells and plants to improve PE employ paired pegRNAs on complementary strands, such as twinPE[41], GRAND[42], PRIME-Del[23], HOPE[43], Bi-PE[44] and DualPE[45]. These strategies enhance editing efficiency and facilitate large fragment edits. In addition, EXPERT[46] introduces an upstream sgRNA (upssgRNA) and an extended pegRNA (ext-pegRNA) at the *cis* strand, enabling higher product purity. Here, FLICK-PE has a single pegRNA but uses two flanking nicking sgRNAs on the non-editing strand, and increases PE efficiency in soybean and tobacco. FLICK-PE could potentially be combined with dual pegRNAs to further improve efficiency and editing capacity. However, the risk of double-stranded breaks may also increase, with potentially four nicks generated on the complementary strands.

Certain canonical strategies could be combined with FLICK-PE for further optimizations of dicot PE tools. For instance, ePE5max[15], PE6[9,47] and PE7 editors[48] could be used over Cas9–MMLV PEmax. Virus replicons, strong promoters and Csy4 recognition elements could be introduced to enhance the expression and stability of PE components. Other optimizations, such as heat-shock treatment and surrogate systems, could also be pyramided to improve PE in dicots.

In gene-editing tool development for plants, transient expression systems (e.g., protoplasts, leaf transient transformation) are widely used for initial tool screening, with subsequent validation via stable transformation[49]. Certain soybean studies also relied on hairy roots as a screening system[20,34,50]. Here, we first optimized soybean PE vectors in hairy roots, then tested their editing efficiency and heritability using stable transformation. Results from three target sites suggest a correlation between the two systems—despite numerical differences in editing efficiency, vectors effective in hairy roots also worked in stable transformation in our hands (Fig. 4). Variability between the two systems is of course unavoidable. Precision editing via PE is complex, and hairy roots differ from stable-transformation systems in cell identity, epigenetic status, and DNA repair environments, all of which may affect efficiency[51–53]. While the hairy-root-to-stable-transformation workflow is well-supported by both prior and current data, the correlation in PE efficiency between the two systems cannot be guaranteed, reflecting their intrinsic differences. In conclusion, for gene-editing tool development in plants, it is practical to use transient systems (or the hairy-root system in the case of soybean) in the initial stages for rapid screening and development; however, subsequent testing and heritability analysis using stable transformation remain essential.

## Methods

### Plant material and growth conditions

Soybean (*Glycine max*) cultivars William 82 (for hairy-root transformation) and Hua-Chun 6 (HC-6, for whole-plant stable transformation) were used, alongside tobacco (*N. benthamiana*). Soybean seedlings for hairy-root transformation and transgenic lines were cultivated under controlled conditions: 400 µmol m$^{-2}$ s$^{-1}$ photosynthetically active radiation, 13 h light (26 °C)/11 h dark (24 °C) photoperiod, and 65% relative humidity. For infiltration, tobacco seeds were germinated and plants were grown at 22 °C under a 16-h light/8 h dark cycle.

Edited soybean plants and wild-type controls were propagated outdoors in Guangzhou, China, during the summer with natural light conditions.

### Design of pegRNAs and nicking sgRNAs

The selection of pegRNA target sites was derived from sgRNAs that had been tested in our research group's previous development of CRISPR–Cas9 technology and demonstrated high knockout efficiency. Additionally, all nicking sgRNAs used in this study were validated to confer high editing activity using gene knockout vectors.

Genomic sequences of target genes were retrieved from the following databases: soybean (*Glycine max Wm*82.*a*6.*v*1) sequences were obtained from Phytozome v13 (https://phytozome-next.jgi.doe.gov), and tobacco (*N. benthamiana NbeHZ*1 *genome* 1.0) sequences were acquired from Nicomics (http://lifenglab.hzau.edu.cn/Nicomics/index.php). The target-site sequences were firstly identified using Sanger sequencing (Tsingke, China) before designing each RTT and PBS. The length of the RTT and PBS in each pegRNA was designed using Plant-PegDesigner v1.0[45] (http://www.plantgenomeediting.net). The non-interfering nucleotide linkers between the pegRNA and 3′ motif (tevopreQ1) were designed via pegLIT[54] (https://peglit.liugroup.us). All design-related information for pegRNAs and nicking sgRNAs in soybean and tobacco is presented in Supplementary Data 1–4.

## Plasmid construction

SoyPE-V1 to SoyPE-V5 vectors were developed through a modular assembly approach[55]. The base vector SoyPE-V1 was engineered from pGES401[20] by replacing *spCas*9 with *nSpCas*9 (H840A) via overlapping PCR[20]. Core components, including *spCas*9n (H840A), *M-MLV* RT (amplified from enpPE2 plasmid[12]), and evopreQ1-HDV (co-amplified from enpPPE2), were assembled into the pOFK entry vector using Golden Gate assembly with *Bsa*I restriction sites (New England Biolabs, R3733L). Subsequent assembly of secondary vectors employing *BsmB*I (New England Biolabs, R0739L) and expression vectors using *Bsa*I completed the SoyPE-V1 construct. This modular strategy was consistently applied across subsequent variants: SoyPE-V2 incorporated a soybean-derived *U*6 composite promoter (PCR-amplified from genomic DNA) replacing the original *Arabidopsis AtU*6 promoter; SoyPE-V3 introduced hyperactive *nSpCas*9 (R211K/N394K/H840A) through overlapping PCR modification; the SoyPE-V2 variant featured the *SV*40-*cMyc* nuclear-localization signal substitution; SoyPE-V4 and SoyPE-V5 respectively used soybean promoters *M*8L and *UBQ*3 in place of *U*6 composite through precision module replacement. SoyPE-V6 was constructed from the SoyPE-V5 backbone through *Pme*I (New England Biolabs, R0560L) digestion and PCR amplification of the *hFTO* expression cassette from pGES501[20] followed by NEBuilder HiFi DNA Assembly Master Mix (New England Biolabs, E2621L).

FLICK-PE-Nt vectors were engineered from the enpPE2 backbone through sequential modifications: (1) dual digestion with *Bsa*I and *Pst*I (New England Biolabs, R0535L and R3140L), (2) PCR amplification of the dual 35*S* promoter from plasmid templates, and (3) seamless assembly using NEBuilder HiFi DNA Assembly Master Mix.

The coding sequences of *GmMLH*1 for *RNAi* (soybean) and *NbMLH*1 for RNAi (tobacco) were amplified by PCR using their respective cDNA as templates. All pegRNA cloning methods for PE vector construction used Golden Gate assembly mediated by *Bsa*I, following type II-S restriction–ligation principles. All PCR amplifications used PrimeSTAR Max DNA Polymerase (Takara, R045A), and sequence verification was conducted by Sanger sequencing (Tsingke, China).

All full-length sequences of the key elements in each vector are listed in Supplementary Data 7.

## Soybean hairy-root transformation

For soybean hairy-root transformation, plasmids were individually introduced into *Agrobacterium rhizogenes* K599 (AngYuBio, China) via the freeze–thaw method. Hypocotyls and cotyledonary nodes from wild-type seedlings were prepared according to a previously described protocol[56]. Briefly, primary roots of 7-day-old seedlings were excised with a slanted cut, and the remaining hypocotyl (retaining a 0.7–1-cm apical segment) was inoculated with *A. rhizogenes* K599 harboring different PE plasmids. Infected seedlings were then transplanted into pots containing sterile wet vermiculite. Transgene-positive roots were screened using a handheld dual-band fluorescence lamp (LUYOR-3415RG, Shanghai Luyor Optoelectronics Co., Ltd., China) at 16 days

post-inoculation, with excitation at 561 nm to activate the DsRED2 fluorescent marker integrated into all PE vectors.

## *Agrobacterium*-mediated tobacco infiltration

For *N. benthamiana* infiltration, plasmids were individually transformed into *A. tumefaciens* GV3101 (AngYuBio, China) via the freeze–thaw method, and six-leaf-stage plants were used for infiltration. Single colonies were first inoculated into 5 mL of LB medium containing 100 mg L$^{-1}$ kanamycin (plasmid selection) and 50 mg L$^{-1}$ rifampicin (strain selection), then incubated overnight at 28 °C with shaking at 200 rpm. Cultures were diluted 1:50 into fresh antibiotic-containing LB medium and grown for an additional 6–8 h at 28 °C until reaching $OD_{600\,nm} = 0.6$–0.8. The suspension was incubated in the dark for 2–3 h at room temperature to induce virulence genes before infiltration. Cells were harvested by centrifugation at $4000 \times g$ for 10 min, resuspended in infiltration buffer (10 mM $MgCl_2$, 10 mM MES monohydrate adjusted to pH 5.7 with KOH, 200 μM acetosyringone), and adjusted to $OD_{600\,nm} = 0.8$. Leaves were infiltrated on the abaxial surface using a needleless 1-mL syringe, and infiltrated tissues were screened using a handheld dual-band fluorescence lamp (LUYOR-3415RG) and collected 3 days post-infiltration for genomic DNA extraction.

## *Agrobacterium*-mediated transformation of soybean and tobacco

For whole-plant transformation of soybean and tobacco, plasmids were individually transformed into *A. tumefaciens* GV3101 (AngYuBio, China) via the freeze–thaw method. Soybean cotyledonary-node transformation followed previously reported protocols[34,57]. In brief, surface-sterilized seeds were germinated, cotyledonary-node explants were inoculated with *Agrobacteria* suspensions, co-cultivated in the dark, and selected on antibiotic-containing medium for shoot regeneration. Resistant shoots were then rooted to generate $T_0$ transgenic plants.

Tobacco leaf-disc transformation was performed using 2–3-week-old *N. benthamiana* according to Horsch et al.[58]. Leaf discs were infected with *A. tumefaciens*, co-cultivated, transferred to hygromycin-containing regeneration medium for shoot induction, and rooted on selective medium to obtain transgenic plants.

## Prime-editing efficiency evaluation and genotyping of transgenic plants

To evaluate the editing efficiency of different PE strategies across the species and respective transformation strategies used in this study, we combined Hi-TOM sequencing with Sanger sequencing[35]. For soybean hairy roots, one positive hairy root was treated as an independent sample with 10–15 samples per biological replicate, while soybean stable-transgenic plants and tobacco used at least three leaves per plant as one sample. Transient tobacco transformation used one infiltrated leaf segment as one sample with 2–3 leaves per replicate.

Genomic DNA was extracted using a cetyltrimethylammonium bromide-based method[59] for use as PCR template for genotyping. Target regions were amplified with site-specific primers (Supplementary Data 5). Editing efficiency was calculated by quantifying intended editing/unintended editing and indel-type reads via Hi-TOM sequencing (e.g., Fig. 2d–j), with effective editing defined as ≥20% intended editing reads[55] (e.g., Fig. 2m). Sanger sequencing validation was performed by Tsingke (China).

## DNA sequencing

For Hi-TOM deep sequencing, genome-specific primers were designed to amplify only 200–300-bp PCR products at the target sites. PCR amplicons were then barcoded using Hi-TOM primers[35] for subsequent sequencing on an Illumina HiSeq X Ten platform. Data analysis was

performed using the Hi-TOM online tool (http://www.hi-tom.net/hi-tom/index-CH.php).

For Sanger sequencing, genome-specific primers were designed to amplify 200–500-bp PCR products covering the target sites. The resulting amplicons were sequenced, and data were analyzed using SnapGene V7.1.2 to identify editing events.

### Analysis of off-target effects

Off-target sites were predicted using CRISPR-GE[60] (http://skl.scau.edu.cn/). For each target, the top-five predicted off-target sites with the highest probability scores were prioritized for analysis by Hi-TOM deep sequencing.

### qRT-PCR of editing-cassette expression

Total RNA was extracted from root tips of transgene-positive hairy roots using the OMEGA Bio-tek Plant RNA Extraction Kit (Catalog No. R6827, OMEGA Bio-tek, USA), following the manufacturer's protocol. Reverse transcription into cDNA was performed using the PrimeScript™ RT reagent Kit (Catalog No. RR037A, Takara Bio Inc., Shiga, Japan) according to the manufacturer's instructions. Quantitative PCR was performed using the SYBR Green PCR Master Mix (Thermo Fisher Scientific, A25742) on a CFX96™ Real-Time PCR Detection System (Bio-Rad, USA) with a 20 μL reaction volume containing 10 μL of 2× SYBR Green PCR Master Mix (final concentration: 1×), 0.8 μL each of 10 μM forward and reverse primers (final concentration: 0.4 μM each), 2 μL of cDNA template (5–10 ng μL$^{-1}$), and 6.4 μL of nuclease-free water. The thermal cycling conditions consisted of pre-denaturation at 95 °C for 3 min, followed by 40 cycles of 95 °C for 10 s (denaturation) and 60 °C for 30 s (annealing/extension), with a subsequent melting-curve analysis at 95 °C for 10 s, 65 °C for 5 s, and a ramp to 95 °C at 0.5 °C s$^{-1}$ to verify amplicon specificity. Soybean translation elongation factor eEF-1α (accession number X56856) was used as a reference to calculate relative expression levels using the $2^{-\Delta CT}$ method[59].

All reactions were performed in technical triplicates for each of the three biological replicates. qRT-PCR primers are listed in Supplementary Data 6.

### Evaluation of glyphosate resistance in edited soybean lines

Homozygous epsps1a$^{TAP-IVS}$ precision-edited soybean lines and wild-type cultivar HC-6 (2-week-old seedlings) were subjected to glyphosate spraying at concentrations of 1.23, 2.46, 4.92, and 9.84 kg a.i. ha$^{-1}$. Treatment solutions were prepared by dissolving 5, 10, 20, and 40 mL of 41% glyphosate isopropylamine salt aqueous concentrate (Nongda 10038784499492, manufactured by Bayer group) in distilled water, respectively, and diluting to 1 L final volume. Seedlings were sprayed until runoff using a handheld pressure sprayer to ensure uniform coverage. Phenotypes were documented 10 days after treatment via photography and visual scoring. Each treatment group comprised 40 individual plants (20 epsps1a$^{TAP-IVS}$ mutants and HC-6 per replicate), with three independent biological replicates (n = 3).

### Statistics and reproducibility

All experiments comparing editing efficiency and phenotypic analyses were performed with at least three independent biological replicates. Technical replicates (e.g., qRT-PCR measurements) were included within each biological replicate as specified in individual assays. No statistical method was used to predetermine sample size. No data were excluded from the analyses. The experiments were not randomized. The investigators were not blinded to allocation during experiments and outcome assessment. For statistical analyses, Microsoft Excel 365 (Microsoft Corp., USA) was used for initial data processing and GraphPad Prism v10.0 (GraphPad Software, USA) was used for statistical analysis and visualization. Tests used, false-

discovery rates and measures of spread are indicated in the respective legends.

### Reporting summary

Further information on research design is available in the Nature Portfolio Reporting Summary linked to this article.

## Data availability

All sequencing data generated in this study have been deposited to the China National Genomics Data Center database under accession code PRJCA041227. The statistic data generated in this study are provided in the Source data file. Source data are provided with this paper.

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

## Acknowledgements

This work was supported by the National Key Research and Development Program of China (2022YFA0912100) to M.B., the Postdoctoral Fellowship Program and China Postdoctoral Science Foundation under Grant Number BX20240094 to M.B., and the Biological Breeding-National Science and Technology Major Project (2023ZD040360104) to Y.G.

## Author contributions

M.B., J.Z., and W.L. performed most experiments, analysed data and contributed to manuscript writing. Y.Z. and H.W. assisted in testing different PE tools in tobacco. M.J., C.P., J.L., and F.H. assisted in vector construction and soybean genotyping. H.K. performed soybean transformation. Y.G. and M.B. designed the project and wrote the paper. Y.G. conducted project administration. All authors have read the manuscript and agree to submission.

## Competing interests

Guangzhou University is listed as the applicant on Chinese patent application No. 2025107482357, with Y.G., M.B., J.Z., and W.L. listed as co-inventors. The remaining authors declare no competing interests.
