## [Transparent Peer Review File · Nature Communications]

A flanking-nicks prime editor (FLICK-PE) system to boost prime editing in dicots

Corresponding Author: Professor Yuefeng Guan

Version 0:

Reviewer comments:

Reviewer #1

(Remarks to the Author)

Comments for the manuscript entitled "Developing a Flanking-Nicks Prime Editor (FLICK-PE) System to Boost Prime Editing in Dicots"

The manuscript presents a dual-nicking strategy called FLICK-PE, which significantly improves the efficiency of prime editing (PE) in soybean and tobacco. The authors successfully demonstrate enhanced editing performance with this system, including heritable edits and practical applications for trait engineering. Overall, the study is methodologically sound and well-documented.

However, I believe the current manuscript needs to address the following points thoroughly before it can be published.

Major points:

1. Use of hairy roots for PE optimization versus cotyledon explants for generating stable edited lines requires clarification.

The optimization of PE tools was conducted in soybean hairy roots, while heritable edits were validated through stable transformation using cotyledonary-node explants. These two systems differ in terms of cell identity, epigenetic status, and DNA repair environments. As a result, the correlation between PE efficiencies in hairy roots and those in stable lines is not guaranteed. This is illustrated by the differences observed at target site 1 in Fig. 3e compared to Fig. 4b. The authors should provide a justification for using hairy roots as a screening platform and discuss any potential variability that may arise.

Although the FLICK-PE approach was tested at multiple locations using the hairy root system, only target site 1 showed successful heritable events via the Agrobacterium-mediated cotyledonary-node transformation. This raises concerns about the effectiveness of the method in generating heritable product events at other target sites. Therefore, I believe it is important to demonstrate that FLICK-PE can produce heritable events at multiple sites (as tested in the hairy root system) in soybean. Additionally, the correlation between the efficiencies of the two systems should be analyzed and discussed.

2. Lack of direct evidence for MMR involvement in the FLICK-PE system.

To support the proposed role of MMR in the FLICK-PE concept, the authors should consider conducting loss-of-function or knockdown experiments targeting MMR components, such as MLH1, in soybean. These experiments would enhance the mechanistic claims and clarify the contribution of MMR to the observed increase in efficiency.

3. Indel and off-target analysis should be expanded.

Although the authors assert that no off-target effects were detected at the chosen loci, their approach may be inadequate. Off-target analysis should be conducted using all the spacers of pegRNAs and the nicking gRNAs at each target site. This is particularly crucial since the FLICK-PE strategy employs three different nicking gRNAs/sites, which resulted in increased double-stranded breaks at on-target sites, as higher indel rates were observed with FLICK-PE at most of the tested sites. Furthermore, this type of analysis should utilize next-generation sequencing (NGS) rather than Sanger sequencing to identify low levels of off-targeting.

The off-targeting analysis should also be conducted with the *Agrobacterium*-mediated cotyledonary-node transformation.

(Optional) A comprehensive profiling of unintended edits, including indel types and potential genome-wide off-target events, would strengthen the claim of editing precision.

Minor points:

1. In my view, comparisons between the tested variants are typically presented using average values, often accompanied by p-values to indicate statistical significance. Occasionally, maximum values are also included. However, in this manuscript, the authors often emphasize maximum values to illustrate the differences between the tested variants.
2. Figure legends should define all abbreviations (e.g., “non-target”, “FLICK-PE-1”, and “FLICK-PE-2”...).
3. Some figures and supplementary figures confuse “Target” and “Traget”.
4. Figure 2e, the font size of the p-value needs to be adjusted.
5. Please ensure to use consistent paired terms between “edited strand”/“non-edited strand” and “editing strand”/“non-editing strand” throughout the text.
6. PE-NESnick is actually PE3. The authors should use one term consistently throughout the text, and I would prefer “PE3”, not a new term.
7. Line #87: What is “canopy-optimization”?
8. Line #116: proU6 or proU6 composite?
9. In Figures 4c-e, the chromatogram for edited line #30 shows a very weak signal for the nucleotide peaks that are expected to be “edited.” However, the inheritance analysis indicated a relatively high allele frequency for this line. I recommend validating the allele frequency in the T0 lines through targeted deep sequencing or by cloning and sequencing the amplified products.
10. Lines #230-232: The claim is misleading because only one data point reaches that level; all other data points show a similar frequency compared to PE3.
11. Lines #233-236: What is the rationale for using the 20% threshold? Is 40.9% and average intended efficiency? And which data refers to the number?

Reviewer #2

(Remarks to the Author)

Manuscript No: NCOMMS-25-46688

Title: Developing a Flanking-Nicks Prime Editor (FLICK-PE) System to Boost Prime Editing in Dicots

Soybean (*Glycine max* L.) is a globally vital staple crop for protein and oil, and is one of the major GMO crops commercialized globally. Prime editing enables installation of base changes and short indels in both mammalian cells and crop plants. However, the prime editing in dicot plants remains challenging. In this study, Bai et al. developed the Flanking-Nicks Prime Editor (FLICK-PE) system by introducing two nick sgRNAs flanking the target site on the non-edited strand. This strategy improved PE efficiency in soybean and tobacco, with up to 35.8-fold and 4.8-fold higher precise editing efficiency than PE2 and PE3, respectively. Using FLICK-PE, the authors efficiently engineered glyphosate resistance by introducing TAP-IVS mutations in EPSPS1a, achieving an intended editing efficiency of 21.1%. The stable edited soybean varieties exhibited vigorous glyphosate tolerance and minimal growth penalties. FLICK-PE also has efficacy in tobacco, underscoring its broad applicability and versatility for rapid, precision breeding in agriculturally important dicot crops. Notably, the development of FLICK-PE will significantly facilitate soybean precision breeding. Given its significance in precision breeding in dicot crops, I would recommend the acceptance of this manuscript for publication in *Nature Communication* after minor revisions.

Minor revisions

1. Suppressing the mismatch repair activity (MMR) will lead to increased prime editing efficacy (Park et al., 2025, *Cell* 188, 1–16). Besides, one of the major merits in this study is that the authors developed the Flanking-Nicks Prime Editor (FLICK-PE) system by introducing two nick sgRNAs flanking the target site on the non-edited strand, which minimized the competition of original DNA non-editing strand for repair, thus significantly increase the prime editing efficiency in soybean and tobacco. I would suggest the authors refine some descriptions in the text in order to avoid confusions.
2. Although the optimized prime editing system such as twinPE or GRAND editor could enable intended insertions or deletions of larger gene fragments in mammalian cells, the scarless insertions of larger fragments in plants remains challenging. Please tone-down some references or descriptions on plant prime editing in the text in order to give the readers a clear and exact view of what is going on and what has been achieved in plant prime editing field.
3. I would like to suggest the authors replace Figure 4I with the Supplementary Fig. 11b for the observation of a more

impressive herbicide tolerant effect of the soybean lines with precisely edited EPSPS1b gene.

4. SoyPE-V6 was constructed from the SoyPE-V5 backbone with the hFTO expression cassette. Does FLICK-PE have the hFTO expression cassette?

5. In Figure 4d, the percentage of different genotypes in the T1 lines is 100.01%, not 100%. Please revise.

6. Some of the median lines in Figure 5c and 5d are missing or not clear.

7. Please provide the full-length sequences of the key elements in each vector in Supplemental Data. This will be helpful for readers to get the useful information.

8. In Supplementary Figure 11, please add the blue arrows or highlight the intended edits in different color.

Version 1:

Reviewer comments:

Reviewer #1

(Remarks to the Author)

Comments for the revised manuscript entitled "A flanking-nicks prime editor (FLICK-PE) system to boost prime editing in Dicots"

The authors have addressed most of my concerns. However, there are still some points to address:

- In major points 1, the authors misunderstood my comment. Explaining why the hairy root system was employed in the initial stage to quickly screen and develop the tools is reasonable. However, what I mean is to discuss the insight and possible potential variability that may arise, somewhere in the discussion part. Moreover, in the rebuttal response, the authors the term "consistent" or "consistency" when compare the two systems, which may not be suitable for the case. In my point of view, "correlated" or "correlation" should be used instead.

- I do not agree with the claim "none of these 105 sites had obvious off-target signals compared to controls." in the rebuttal response or "no off-target effect was detected (Supplementary Figs. 8–14)" in lines #246-247 of the revised manuscript because some potential off-target sites showed up to 10% of Indel/SNPs sequencing reads (e.g. Off-T1 of pegRNA 1 in Supplementary Fig. 8) and many OT sites showed ~5% of Indel/SNPs sequencing reads, while the control showed nearly 0%. The claim in lines #285-286, "no off-target editing at the potential off-target sites (Supplementary Fig. 16)" is also not reasonable since in Supplementary Fig.16, some OT sites showed higher off-target activity compared to the controls. The authors should change the conclusions regarding the off-target analysis discussion.

- Also, there are many Supplementary Fig. 9s but omitted Supplementary Fig. 10-14.

Reviewer #2

(Remarks to the Author)

In the revised manuscript, the authors have addressed all my concerns. Given the considerable significance of this work for advancing precision breeding in dicot crops, as well as the rigorous supplementary experiments conducted, I recommend its acceptance for publication in Nature Communications after minor revisions for clarity.

Minor points

1 Line 22-24 This description may cause misunderstanding and could be omitted.

2 Line 75–77, 79–80, 204–206, 503–505, 544–555 these descriptions may not be conclusive and should be revised or deleted.

3 The contents in Lines 299–308 in the tracked manuscript file is missing.

4 Some words/sentences in the manuscript need further language improvements or minor revisions, e.g., "Dicots" in the title needs no capitalization; "TAP-IVS" in the key words could be revised into EPSPS; Line 52 (reference citation), Line 58 "markedly" should be omitted; Lines 157–158 "this by manipulating MMR to check" could be omitted; Line 170 "nick sgRNA" should be corrected to "nicking sgRNA"; Line 196 "In all efficient events with nicks...", this sentence is unclear; Line 200 "intended-editing events" needs revision; Line 240 "saw" is improper; Lines 492–493, 'reaches 18.3%, or 19.4%...', please add 'respectively'.

**REVIEWER COMMENTS**

**Reviewer #1 (Remarks to the Author):**

The manuscript presents a dual-nicking strategy called FLICK-PE, which significantly
improves the efficiency of prime editing (PE) in soybean and tobacco. The authors
successfully demonstrate enhanced editing performance with this system, including
heritable edits and practical applications for trait engineering. Overall, the study is
methodologically sound and well-documented.

However, I believe the current manuscript needs to address the following points
thoroughly before it can be published.

Major points:

1. Use of hairy roots for PE optimization versus cotyledon explants for generating stable
edited lines requires clarification.

The optimization of PE tools was conducted in soybean hairy roots, while heritable
edits were validated through stable transformation using cotyledonary-node explants.
These two systems differ in terms of cell identity, epigenetic status, and DNA repair
environments. As a result, the correlation between PE efficiencies in hairy roots and
those in stable lines is not guaranteed. This is illustrated by the differences observed at
target site 1 in Fig. 3e compared to Fig. 4b. The authors should provide a justification
for using hairy roots as a screening platform and discuss any potential variability that
may arise.

**Response:**

Thank you for the constructive comments. For technical optimization, the
*Agrobacterium rhizogenes*-mediated hairy-root assay serves as a rapid and powerful
screening platform, primarily due to speed, efficiency, scalability, and the generally
consistent editing efficiency compared to stable transformation.

Stable transformation (i.e. *Agrobacterium tumefaciens*-mediated cotyledon-node
transformation) in soybean and many crops is generally slow (4 months in soybean),
technically recalcitrant (5–10% transformation efficiency in soybean), and expensive.
In comparison, *Agrobacterium rhizogenes*-based transformation of soybean roots to
form composite plants takes only 2–3 weeks, is highly efficient (50–100%
transformation frequency), and is cost effective. Hairy roots have been widely used as
a powerful tool in gene functional studies (*Ke et al, Science, 2022*¹; *Yu et al, The Plant*
*Cell, 2023*²). and development of gene-editing tools, particularly in legume species
(*Zhao et al, The Crop J. 2024*³; *Zhang et al, Front. Plant Sci., 2020*⁴, *Alamillo et al,*
*Curr. Opin. Biotech. 2023*⁵; *Shu et al, BMC Plant Biol. 2020*⁶).

Likewise, transient transformation systems (e.g., protoplasts, leaf transient
transformation) were widely adopted technical approaches in plant gene editing (*Gao*
*et al, Cell, 2021*⁷). For example, the first establishment of PE in plants involved

evaluating the efficiency of various optimized vectors via rice and wheat protoplasts,
followed by screening efficient vectors for testing in rice stable transformation (*Lin et al, Nat Biotechnol*, 2020⁸); the development of PE tools in maize (*Jiang et al, Plant*
*Biotechnol J*, 2022⁹) and tobacco (*Zhao et al, Nat Plants*, 2024¹⁰) also follows this
approach.

Before this study, the development of soybean PE tools had not been reported, but
in the development of soybean mutagenesis tools (*Bai et.al , Mol Plant*, 2024¹¹) and
base-editing tools (*Huang et al, Cell*, 2023¹²), the soybean hairy-root system was used
as a vector-screening platform in all cases. In addition, compared with transient
transformation systems, hairy roots present numerous advantages— in gene function
studies with legumes, this system is often used to replace stable transformation for
functional verification of gene over-expression or knockout. Therefore, using the
soybean hairy roots for gene-editing vector testing is a reliable and widely recognized
technical strategy.

In terms of the overall correlation of editing efficiency, there is consistency
between soybean stable transformation and the hairy-root system. For example, we
found in previous studies that when developing soybean gene-knockout tools, screening
sgRNAs with high activity via in hairy roots first and then applying them to stable
transformation can improve gene-editing efficiency (*Bai et.al, Plant Biotechnol J*,
2020¹³). Meanwhile, in response to the suggestion, two additional target sites were
selected for prime-editing verification via stable transformation in the revised
manuscript (**Fig. 4**). From the three tested target sites, vectors that can achieve effective
editing in the hairy-root system also seem to exert effective editing outcomes in the
upon stable transformation, and although there are differences in the specific values of
editing efficiency, these are sufficient to reflect the core consistency between the two
systems. In addition, the efficiency of soybean stable transformation is extremely low,
requiring substantial human and material resources. Therefore, in the development of
soybean PE tools, prioritizing the screening and verification of PE tools via the hairy-
root system, followed by further, robust validation of efficiency and heritability via
stable transformation, constitutes a workflow that combines feasibility and cost-
effectiveness.

**References**

- 1. Ke, X. et al. Phosphoenolpyruvate reallocation links nitrogen fixation rates to
root nodule energy state. *Science* **378**, 971-977 (2022).
- 2. Yu, H. et al. GmNAC039 and GmNAC018 activate the expression of cysteine
protease genes to promote soybean nodule senescence. *Plant Cell* **35**, 2929-
2951 (2023).
- 3. Zhao, H. et al. Development of a single transcript CRISPR/Cas9 toolkit for

- efficient genome editing in autotetraploid alfalfa. *The Crop Journal* **12**, 788-795
(2024).
- 4. Zhang, H. et al. Efficient Generation of CRISPR/Cas9-Mediated
Homozygous/Biallelic *Medicago truncatula* Mutants Using a Hairy Root
System. *Front Plant Sci* **11**, 294 (2020).
- 5. Alamillo, J.M. et al. Clustered regularly interspaced short palindromic
repeats/CRISPR-associated protein and hairy roots: a perfect match for gene
functional analysis and crop improvement. *Curr Opin Biotechnol* **79**, 102876
(2023).
- 6. Shu, H., Luo, Z., Peng, Z. & Wang, J. The application of CRISPR/Cas9 in hairy
roots to explore the functions of *AhNFR1* and *AhNFR5* genes during peanut
nodulation. *BMC Plant Biol* **20**, 417 (2020).
- 7. Gao, C. Genome engineering for crop improvement and future agriculture. *Cell*
**184**, 1621-1635 (2021).
- 8. Lin, Q. et al. Prime genome editing in rice and wheat. *Nat Biotechnol* **38**, 582-
585 (2020).
- 9. Jiang, Y.Y. et al. Prime editing efficiently generates W542L and S621I double
mutations in two *ALS* genes in maize. *Genome Biol* **21**, 257 (2020).
- 10. Zhao, Y. et al. Precise deletion, replacement and inversion of large DNA
fragments in plants using dual prime editing. *Nat Plants* **11**, 191-205 (2025).
- 11. Bai, M. et al. Expressing a human RNA demethylase as an assister improves
gene-editing efficiency in plants. *Mol Plant* **17**, 363-366 (2024).
- 12. Huang, J. et al. Discovery of deaminase functions by structure-based protein
clustering. *Cell* **186**, 3182-3195 e3114 (2023).
- 13. Bai, M. et al. Generation of a multiplex mutagenesis population via pooled
CRISPR-Cas9 in soya bean. *Plant Biotechnol J* **18**, 721-731 (2020).

Although the FLICK-PE approach was tested at multiple locations using the hairy root
system, only target site 1 showed successful heritable events via the Agrobacterium-
mediated cotyledonary-node transformation. This raises concerns about the
effectiveness of the method in generating heritable product events at other target sites.
Therefore, I believe it is important to demonstrate that FLICK-PE can produce heritable
events at multiple sites (as tested in the hairy root system) in soybean. Additionally, the
correlation between the efficiencies of the two systems should be analyzed and
discussed.

**Response:**

We agree, and to demonstrate the effectiveness of FLICK-PE, we have added stable-
transformation data for Target-sites 3 and 5 using PE2, PE3 and FLICK-PE (**Fig. 4**).
For the two additional targets, FLICK-PE can achieve editing efficiencies of 11.1% at

Target 3 (3/27 lines) which was higher than the 6.7% (2/30 lines) for PE3. Consistent
results were observed at Target-site 5, FLICK-PE achieved an effective intended-
editing efficiency of 16% (4/25), compared to 3.1% (1/32) for PE3 (**Fig. 4c**).
Furthermore, PE2 failed to generate effective editing at any of the tested target sites in
stable transformation. Collectively, FLICK-PE enables effective intended editing in
soybean via stable transformation, with an efficiency range of 11.1–21.1% (mean
16.1%), higher than that of PE3 (mean intended editing efficiency: 6.2%, range: 3.1–
8.7%), representing an average 2.6-fold increase (Lines 263-283).

In addition, we have also stated that the soybean hairy-root results are consistent
with those obtained via stable transformation in the Discussion section (Lines 355-359).

2.Lack of direct evidence for MMR involvement in the FLICK-PE system.

To support the proposed role of MMR in the FLICK-PE concept, the authors should
consider conducting loss-of-function or knockdown experiments targeting MMR
components, such as *MLH1*, in soybean. These experiments would enhance the
mechanistic claims and clarify the contribution of MMR to the observed increase in
efficiency.

**Response:**

In the revised manuscript, we constructed PE vectors co-expressing an RNAi cassette
against *MLH1* (**Supplementary Fig. 15**). Hairy-root experiments (Target-sites 1 and 3)
showed that efficiency increased significantly upon *MLH1* knock-down compared to
PE2/PE3 controls lacking this cassette. Surprisingly, FLICK-PE + *MLH1* RNAi showed
no obvious efficiency difference from FLICK-PE. However, FLICK-PE without MMR
inhibition showed higher editing efficiency than PE2/PE3 + *MLH1* RNAi.

Both nicking sgRNAs and *MLH1* RNA interference may work through inhibiting
the possibility of repair of the precise-editing products. Thus, FLICK-PE with two
nicking sgRNAs might override the effect of *MLH1* knock-down. Considering that
*MLH1* RNA interference causes defects in stable-transgenic rice, FLICK-PE represents
a more feasible and effective approach in soybean than *MLH1* RNAi. We have added
corresponding results and discussion text (Lines 248-258 and 378-384).

3.Indel and off-target analysis should be expanded.

Although the authors assert that no off-target effects were detected at the chosen loci,
their approach may be inadequate. Off-target analysis should be conducted using all the
spacers of pegRNAs and the nicking gRNAs at each target site. This is particularly
crucial since the FLICK-PE strategy employs three different nicking gRNAs/sites,

which resulted in increased double-stranded breaks at on-target sites, as higher indel
rates were observed with FLICK-PE at most of the tested sites. Furthermore, this type
of analysis should utilize next-generation sequencing (NGS) rather than Sanger
sequencing to identify low levels of off-targeting.

The off-targeting analysis should also be conducted with the *Agrobacterium*-mediated
cotyledonary-node transformation.

(Optional) A comprehensive profiling of unintended edits, including indel types and
potential genome-wide off-target events, would strengthen the claim of editing
precision.

**Response:**

Following these suggestions, we have conducted a comprehensive evaluation of off-
target effects of the FLICK-PE strategy in the revised manuscript:

- • We detected potential off-target sites (105 in total) corresponding to all
pegRNAs (7) and nicking sgRNAs (14) across all tested target sites using Hi-
TOM deep sequencing, with negative hairy-root DNA samples as controls. As
shown in **Supplementary Figs. 8–14**, none of these 105 sites had obvious off-
target signals compared to controls.
- • Additionally, in T₀ lines generated via *Agrobacterium*-mediated cotyledonary-
node stable transformation, we evaluated 45 potential off-target sites
corresponding to all pegRNAs and nicking sgRNAs of three tested target sites
using Hi-TOM deep sequencing. No obvious off-target signals were detected at
these 45 sites (**Supplementary Fig. 16**).
- • In the revised manuscript, we supplemented the rate of unintended edits and
indels in all targets.
- • For some data in the main figures: intended edits, unintended edits, and indels
are presented in the same figure (e.g., **Fig. 3e–i**); while other relevant data are
provided in the supplementary data, with relevant data from Fig. 2 included in
Supplementary Fig. 4, and relevant data from Fig. 4b,c are included in
Supplementary Fig. 6.

**Minor points:**

1. In my view, comparisons between the tested variants are typically presented using
average values, often accompanied by p-values to indicate statistical significance.
Occasionally, maximum values are also included. However, in this manuscript, the
authors often emphasize maximum values to illustrate the differences between the
tested variants.

**Response:**

We have revised the entire text to give a more measured assessment by 1)

supplementing average editing-efficiency data for each target site in all efficiency-
comparison sections (e.g., PE2/PE3, FLICK-PE analysis); 2) reduced emphasis on
"maximum values" with accompanying average values and statistical results.

2. Figure legends should define all abbreviations (e.g., "non-target", "FLICK-PE-1",
and "FLICK-PE-2"...).

**Response:**

Done.

3. Some figures and supplementary figures confuse "Target" and "Traget".

**Response:**

Addressed.

4. Figure 2e, the font size of the p-value needs to be adjusted.

**Response:**

Addressed.

5. Please ensure to use consistent paired terms between "edited strand"/"non-edited
strand" and "editing strand"/"non-editing strand" throughout the text.

**Response:**

Addressed.

6. PE-NESnick is actually PE3. The authors should use one term consistently throughout
the text, and I would prefer "PE3", not a new term.

**Response:**

Changed to 'PE3' throughout.

7. Line #87: What is "canopy-optimization"?

**Response:**

We have revised this to "canonical optimization".

8. Line #116: proU6 or proU6 composite?

**Response:**

We have revised "proU6" to "*proU6* composite".

9. In Figures 4c-e, the chromatogram for edited line #30 shows a very weak signal for
the nucleotide peaks that are expected to be "edited." However, the inheritance analysis
indicated a relatively high allele frequency for this line. I recommend validating the
allele frequency in the T0 lines through targeted deep sequencing or by cloning and

sequencing the amplified products.

**Response:**

Regarding the detection of T₀ lines, we first analyzed them via Hi-TOM sequencing
and the relevant results are presented in **Fig. 4b**; the Sanger sequencing results in **Fig.**
**4c**, meanwhile, served as a secondary validation of the Hi-TOM results. In the revised
manuscript, we have added the "intended edit reads" (derived from targeted deep
sequencing) alongside the Sanger sequencing chromatograms for this line.

10.Lines #230-232: The claim is misleading because only one data point reaches that
level; all other data points show a similar frequency compared to PE3.

**Response:**

We have revised this to avoid overstatement (Lines 235-236).

11.Lines #233-236: What is the rationale for using the 20% threshold? Is 40.9% and
average intended efficiency? And which data refers to the number?

**Response:**

Our criterion for "effective intended editing " is the editing events that are most likely
stably inherited to the next generation. Based on this, it is necessary to quantify effective
editing by setting a intended-editing reads proportion threshold from Hi-TOM deep
sequencing. In the soybean stable-transformation experiment using FLICK-PE, we
observed that intended-editing events with a Hi-TOM read proportion of > 20% could
be a threshold achieved stable inheritance (e.g., transformant #30 in Fig. 4c, which had
a intended editing proportion of 24.36% in the T₀ generation, and the inheritance of this
editing was successfully detected in the T₁ generation, see **Fig. 5b** for details).

Regarding the 40.9% effective intended-editing efficiency at Target-Site 1, this
value is the statistical average of three biological replicates. According to the data
shown in **Fig. 3e**, with FLICK-PE, the total number of positive hairy roots across the
three replicates was 22, among which 9 hairy roots met the "effective intended-editing
threshold (>20%)". The effective-editing efficiency was calculated as "number of
effective hairy roots / total number of positive hairy roots" ($9/22 \times 100\% \approx 40.9\%$).
This result is also supplemented in **Fig. 3k** to ensure traceability.

**Reviewer #2 (Remarks to the Author):**

Manuscript No: NCOMMS-25-46688

Title: Developing a Flanking-Nicks Prime Editor (FLICK-PE) System to Boost Prime
Editing in Dicots

Soybean (*Glycine max* L.) is a globally vital staple crop for protein and oil, and is one
of the major GMO crops commercialized globally. Prime editing enables installation of
base changes and short indels in both mammalian cells and crop plants. However, the

prime editing in dicot plants remains challenging. In this study, Bai et al. developed the
Flanking-Nicks Prime Editor (FLICK-PE) system by introducing two nick sgRNAs
flanking the target site on the non-edited strand. This strategy improved PE efficiency
in soybean and tobacco, with up to 35.8-fold and 4.8-fold higher precise editing
efficiency than PE2 and PE3, respectively. Using FLICK-PE, the authors efficiently
engineered glyphosate resistance by introducing TAP-IVS mutations in EPSPS1a,
achieving an intended editing efficiency of 21.1%. The stable edited soybean varieties
exhibited vigorous glyphosate tolerance and minimal growth penalties. FLICK-PE also
has efficacy in tobacco, underscoring its broad applicability and versatility for rapid,
precision breeding in agriculturally important dicot crops. Notably, the development of
FLICK-PE will significantly facilitate soybean precision breeding. Given its
significance in precision breeding in dicot crops, I would recommend the acceptance of
this manuscript for publication in Nature Communication after minor revisions.

Minor revisions

1. Suppressing the mismatch repair activity (MMR) will lead to increased prime editing
efficacy (Park et al., 2025, Cell 188, 1–16). Besides, one of the major merits in this
study is that the authors developed the Flanking-Nicks Prime Editor (FLICK-PE)
system by introducing two nick sgRNAs flanking the target site on the non-edited strand,
which minimized the competition of original DNA non-editing strand for repair, thus
significantly increase the prime editing efficiency in soybean and tobacco. I would
suggest the authors refine some descriptions in the text in order to avoid confusions.

**Response:**

Thank you for your comments. In the revised manuscript, we constructed PE vectors
co-expressing an RNAi cassette against *MLH1* (**Supplementary Fig. 15**). Indeed,
soybean hairy-root experiments (Target-sites 1, 3) showed that PE2 + *MLH1* RNAi and
PE3 + *MLH1* RNAi efficiency increased compared to PE2/PE3 controls. Surprisingly,
FLICK-PE + *MLH1* RNAi showed no difference in efficiency from FLICK-PE.
However, FLICK-PE without MMR inhibition showed higher editing efficiency than
PE2/PE3 + *MLH1* RNAi .

Both nicking sgRNAs and *MLH1* RNA interference may work through inhibiting
the possibility of repair of the precise-editing products. Thus, FLICK-PE with two
nicking sgRNAs might override the effect of *MLH1* knock-down. Considering that
*MLH1* RNA interference cause defects in stable-transgenic rice, FLICK-PE represents
a more feasible and effective approach in soybean than *MLH1* RNAi. We have added
corresponding results and discussion text (Lines 248-258 and 378-384).

.

2. Although the optimized prime editing system such as twinPE or GRAND editor could
enable intended insertions or deletions of larger gene fragments in mammalian cells,

the scarless insertions of larger fragments in plants remains challenging. Please tone-
down some references or descriptions on plant prime editing in the text in order to give
the readers a clear and exact view of what is going on and what has been achieved in
plant prime editing field.

**Response:**

Done.

3.I would like to suggest the authors replace Figure 4I with the Supplementary Fig. 11b
for the observation of a more impressive herbicide tolerant effect of the soybean lines
with precisely edited EPSPS1b gene.

**Response:**

Done.

4.SoyPE-V6 was constructed from the SoyPE-V5 backbone with the hFTO expression
cassette. Does FLICK-PE have the hFTO expression cassette?

**Response:**

Since SoyPE-V6 (constructed from the SoyPE-V5 backbone with the *hFTO* cassette)
showed no effect in hairy roots and stable transformation, we did not incorporate the
*hFTO* expression cassette into FLICK-PE.

5.In Figure 4d, the percentage of different genotypes in the T1 lines is 100.01%, not
100%. Please revise.

**Response:**

This discrepancy resulted from rounding errors and we have revised the percentages to
ensure their total equals exactly 100%. This is addressed in revised **Fig. 5b**.

6.Some of the median lines in Figure 5c and 5d are missing or not clear.

**Response:**

Addressed.

7.Please provide the full-length sequences of the key elements in each vector in
Supplemental Data. This will be helpful for readers to get the useful information.

**Response:**

We have included all these sequences in **Supplemental Data 7** as suggested.

8.In Supplementary Figure 11, please add the blue arrows or highlight the intended edits
in different color.

**Response:**

Done.

REVIEWER COMMENTS

Reviewer #1 (Remarks to the Author):

Comments for the revised manuscript entitled “A flanking-nicks prime editor (FLICK-PE) system to boost prime editing in Dicots”

The authors have addressed most of my concerns. However, there are still some points to address:

- In major points 1, the authors misunderstood my comment. Explaining why the hairy root system was employed in the initial stage to quickly screen and develop the tools is reasonable. However, what I mean is to discuss the insight and possible potential variability that may arise, somewhere in the discussion part. Moreover, in the rebuttal response, the authors the term “consistent” or “consistency” when compare the two systems, which may not be suitable for the case. In my point of view, “correlated” or “correlation” should be used instead.

Response:

We regret the misunderstanding here. In this revision, we have replaced all instances of "consistent" or "consistency" with "correlated" or "correlation" throughout the text. we also added relevant descriptions in the Discussion section as suggested as follows: “In gene-editing tool development for plants, transient expression systems (e.g., protoplasts, leaf transient transformation) are widely used for initial tool screening, with subsequent validation via stable transformation⁴⁹. Certain soybean studies also relied on hairy roots as a screening system^{20, 34, 50}. Here, we first optimized soybean PE vectors in hairy roots, then tested their editing efficiency and heritability using stable transformation. Results from three target sites suggest a correlation between the two systems — despite numerical differences in editing efficiency, vectors effective in hairy roots also worked in stable transformation in our hands (Fig. 4). Precision editing via PE is complex, and hairy roots differ from stable-transformation systems in cell identity, epigenetic status, and DNA repair environments, all of which may affect efficiency⁵¹⁻⁵³. In conclusion, for gene-editing tool development in plants, it is practical to use transient systems (or the hairy-root system in the case of soybean) in initial stages for rapid screening and development; however, subsequent testing and heritability analysis using stable transformation remain essential.”(Lines 402–415).

- I do not agree with the claim “none of these 105 sites had obvious off-target signals compared to controls.” in the rebuttal response or “no off-target effect was detected (Supplementary Figs. 8–14)” in lines #246-247 of the revised manuscript because some potential off-target sites showed up to 10% of Indel/SNPs sequencing reads (e.g. Off-T1 of pegRNA 1 in Supplementary Fig. 8) and many OT sites showed ~5% of Indel/SNPs sequencing reads, while the control showed nearly 0%. The claim in lines #285-286, “no off-target editing at the potential off-target sites (Supplementary Fig. 16)” is also not reasonable since in Supplementary Fig.16, some OT sites showed higher off-target activity compared to the controls. The authors should change the conclusions regarding the off-target analysis discussion.

Response:

We appreciate the careful observation around off-target analysis. The conclusions related to off-target analysis have been revised: we have replaced the previous claim of "no obvious off-target signals" with a statement that "some off-target events were observed at potential off-target sites". Importantly, all these off-target events did not exceed the 20% threshold for heritable effective editing. We now overtly conclude that a low ratio of off-target risk may exist at different target sites (**Lines 240-246 and 285–291**).

- Also, there are many Supplementary Fig. 9s but omitted Supplementary Fig. 10-14.

Response:

Addressed.

Reviewer #2 (Remarks to the Author):

In the revised manuscript, the authors have addressed all my concerns. Given the considerable significance of this work for advancing precision breeding in dicot crops, as well as the rigorous supplementary experiments conducted, I recommend its acceptance for publication in Nature Communications after minor revisions for clarity.

Minor points

1 Line 22-24 This description may cause misunderstanding and could be omitted.

Response:

Done.

2 Line 75–77, 79–80, 204–206, 503–505, 544–555 these descriptions may not be conclusive and should be revised or deleted.

Response:

Addressed.

3 The contents in Lines 299–308 in the tracked manuscript file is missing.

Response:

Addressed.

4 Some words/sentences in the manuscript need further language improvements or minor revisions, e.g., "Dicots" in the title needs no capitalization; "TAP-IVS " in the key words could be revised into EPSPS ; Line 52 (reference citation), Line 58 "markedly" should be omitted; Lines 157–158 "this by manipulating MMR to check" could be omitted; Line 170 "nick sgRNA" should be corrected to "nicking sgRNA"; Line 196 "In all efficient events with nicks..." , this sentence is unclear; Line 200 "intended-editing events" needs revision; Line 240 "saw" is improper; Lines 492–493, 'reaches 18.3%, or 19.4%...' , please add 'respectively'.

Response:

All addressed.